# EXPLOITING PLAYBACKS IN UNSUPERVISED DOMAIN ADAPTATION FOR 3D OBJECT DETECTION

## ABSTRACT

Self-driving cars must detect other vehicles and pedestrians in 3D to plan safe routes and avoid collisions. State-of-the-art 3D object detectors, based on deep learning, have shown promising accuracy but are prone to over-fit to domain idiosyncrasies, causing them to fail in new environments—a serious problem if autonomous vehicles are meant to operate freely. In this paper, we propose a novel learning approach that drastically reduces this gap by fine-tuning the detector on pseudo-labels in the target domain, which our method generates while the vehicle is parked, based on replays of previously recorded driving sequences. In these replays objects are tracked over time and detections are interpolated and extrapolated—crucially, leveraging future information to catch hard cases. We show, on five autonomous driving datasets, that fine-tuning the detector on these pseudo-labels substantially reduces the domain-gap to new driving environments, yielding drastic improvements in accuracy and detection reliability.

## 1 INTRODUCTION

One of the fundamental learning problems in the context of self-driving cars, is the detection and localization of other traffic participants, such as cars, cyclists, and pedestrians in 3D. Typically, the input consists of LiDAR or pseudo-LiDAR (Wang et al., 2019b) point clouds (sometimes with accompanying images), and the outputs are sets of tight 3D bounding boxes that envelope the detected objects. The problem is particularly challenging, because the predictions must be highly accurate, reliable, and, importantly, be made in real time. The current state-of-the-art in 3D object detection is based on deep learning approaches (Qi et al., 2018; Shi et al., 2019; Yang et al., 2018; Shi et al., 2020), trained on short driving segments with labeled bounding boxes (Geiger et al., 2012; 2013), which yield up to $80\%$ average precision on held-out segments (Shi et al., 2020).

However, as with all machine learning, these techniques succeed when the training data distribution matches the test data distribution. One possibility to ensure train/test consistency is to constrain self-driving cars to a small geo-fenced area. Here, a fleet of self-driving taxis might together collect accurate training data with exhaustive coverage so that the accuracy of the system is guaranteed. This, however, is fundamentally limiting. Ultimately, one would like to allow self-driving cars to be driven freely anywhere, similar to a human driven car. This unconstrained scenario introduces an inherent adaptation problem: The car producer cannot foresee where the owner will ultimately operate the car. The perception system might be trained on urban roads in Germany (Geiger et al., 2013; 2012), but the car may be driven in the mountain regions in the USA, where other cars may be larger and fewer, the roads may be snowy, and buildings may look different. Past work has shown that such differences can cause $>35\%$ drop in the accuracy of extant systems (Wang et al., 2020). Closing this adaptation gap is one of the biggest remaining challenges for freely self-driving vehicles.

Car owners, however, are likely to spend most of their driving time on similar routes (commuting to work, grocery stores, etc.), and leave their cars parked (*e.g.*, at night) for extended amounts of time. This raises an intriguing possibility: the car can collect training data on these frequent trips; then retrain itself while offline to adapt to this new environment for subsequent online driving. Unfortunately, the data the car collects are *unlabeled*. The challenge is thus in *unsupervised domain adaptation* (Gong et al., 2012): The detection system, having been previously trained on labeled data from a *source* domain, must now adapt to a *target* domain where only unlabeled data are available.

In this paper, we present a novel and effective approach for unsupervised domain adaptation of 3D detectors that addresses this challenge. Our key insight is two-fold. One, our data is not simply a bag of independent images, but a *video* of the same scene over time. Two, the *dynamics* of our objects of interest (*i.e.*, cars) can be modeled effectively. This allows us to take confident detections of nearby objects, estimate their states (*e.g.*, locations, sizes and speeds), and then *extrapolate* them forward and backward in time, when they were missed by the detector. We show that these extrapolations allow us to correctly classify *precisely* the difficult (typically distant) objects that are easily missed in new environments. For example, an oncoming car may be detected too late, only when it is close enough. The playback allows us to go back in time and annotate its position in frames where it was previously missed. Although this process cannot be performed in real time (since it uses future information), we use it to generate a new training set with pseudo-labels for the target environment. We then adapt the detector to the target domain through fine-tuning on this newly created data set. The few, but likely accurate labels allow the detector to generalize to more settings in this environment (*e.g.*, picking up what distant cars or typical background scenes look like). We call our approach *dreaming*, as the car learns by replaying past driving sequences backwards and forwards while it is parked.

We evaluate our dreaming car algorithm on multiple autonomous driving datasets, including KITTI (Geiger et al., 2012; 2013), Argoverse (Chang et al., 2019), Lyft (Kesten et al., 2019), Waymo (Sun et al., 2019), and nuScenes (Caesar et al., 2019). We show across all possible data set combinations that fine-tuning the detector with our dreaming car procedure drastically reduces the source/target domain gap with high consistency. In fact, the resulting detector after "dreaming" *substantially exceeds* the accuracy of the offline system used to generate the pseudo-labels, which —although able to look into the future— is limited to the extrapolation of confident detection before adaption. Our dreaming procedure can easily be implemented on-device and we believe that it constitutes a significant step towards safely operating autonomous vehicles without geo-restrictions.

## 2 RELATED WORK

**3D object detection.** Prior work can be categorized based on the input sensors: using 3D sensors (time-of-flight sensors) like Light Detection and Ranging (LiDAR) or 2D images from inexpensive commodity cameras (Wang et al., 2019b; You et al., 2019; Qian et al., 2020; Li et al., 2019; Chen et al., 2020). We focus on LiDAR-based methods due to their higher accuracy. LiDAR-based 3D object detectors view the LiDAR signal as a 3D point cloud. For example, Frustum PointNet (Qi et al., 2018) applies PointNet (Qi et al., 2017a;b), a neural network dedicated to point clouds, to LiDAR points within each image-based frustum proposal to localize the object in 3D. PointRCNN (Shi et al., 2019) combines PointNet with faster R-CNN (Ren et al., 2015) to directly generate proposals in 3D using LiDAR points alone. VoxelNet (Zhou & Tuzel, 2018) and PointPillar (Lang et al., 2019) encode 3D points into voxels and extract features by 3D convolutions and PointNet. For scenes dedicated to self-driving cars, processing points from the top-down bird's-eye view (BEV) also proves to be sufficient for capturing object contours and locations (Ku et al., 2018; Yang et al., 2018; Liang et al., 2018). While all these algorithms have consistently improved the detection accuracy, they are mainly evaluated on KITTI (Geiger et al., 2012) alone. Wang et al. (2020) recently revealed the poor generalization of 3D detectors when they are trained and tested on different datasets, especially on distant objects with sparse LiDAR points.

**Unsupervised domain adaptation (UDA).** UDA has been widely studied in the machine learning and computer vision communities, especially on image classification (Gopalan et al., 2011; Gong et al., 2012; Ganin et al., 2016; Tzeng et al., 2017; Saito et al., 2018). The common setup is to adapt a model trained from one labeled source domain (*e.g.*, synthetic images) to another unlabeled target domain (*e.g.*, real images). Recent work has extended UDA to driving scenes, but mainly for 2D semantic segmentation and 2D object detection. (See Appendix A for a list of work.) The mainstream approach is to match the feature distributions or image appearances between domains, for example, via adversarial learning (Ganin et al., 2016; Hoffman et al., 2018; Tzeng et al., 2017) or image translation (Zhu et al., 2017). The approaches more similar to ours are (RoyChowdhury et al., 2019; Tao et al., 2018; Liang et al., 2019; Zhang et al., 2018; Zou et al., 2018; Choi et al., 2019; Chitta et al., 2018; Yu et al., 2019; Kim et al., 2019a; French et al., 2018; Inoue et al., 2018; Rodriguez & Mikolajczyk, 2019; Khodabandeh et al., 2019; Chen et al., 2011), which iteratively assign pseudo-labels to (some of) the target domain unlabeled data and re-train the models. This procedure, usually named self-training, has proven to be effective in learning with unlabeled data,

such as semi-supervised and weakly-supervised learning (McClosky et al., 2006b;a; Kumar et al., 2020; Lee, 2013; Cinbis et al., 2016; Triguero et al., 2015). Specifically for UDA, self-training enables the model to adapt its features to the target domain in a supervised learning fashion.

*For UDA in 3D, the domain discrepancy is in the point clouds, instead of images.* Qin et al. (2019) are the first to map and match point clouds between domains, via adversarial learning. However, the point clouds they considered correspond to single, isolated objects, as opposed to the cluttered, large scenes encountered in driving, which are considerably more challenging. Others project LiDAR points to a frontal or BEV and apply UDA techniques in the resulting 2D images for BEV object detection (Saleh et al., 2019; Wang et al., 2019c) or semantic segmentation (Wu et al., 2019). However, this approach forces the downstream detector to operate on the 2D image, which can be sub-optimal. There is a need for UDA techniques that apply to large point clouds from unstructured, cluttered scenes.

Our work is the first to apply self-training for UDA to realistic scenes for 3D object detection. We argue that this can be more effective than learning a mapping from the target to the source, for which the detector is (over-)specialized. First, a LiDAR point cloud easily contains more than $10,000$ points, making full-scene transformations prohibitively slow. Second, in many practical cases, we may not have access to source domain training data (Chidlovskii et al., 2016) after the detector is deployed— making it impossible to learn a cross-domain mapping. While most self-training approaches assume access to source domain data, we make no such assumption and never use it in the adaptation process.

**Leveraging videos for object detection.** To ease the annotation efforts for 2D object detection, several works (Liang et al., 2015; Ošep et al., 2019; Misra et al., 2015; Kumar Singh et al., 2016) proposed to mine additional bounding boxes from videos in an unsupervised or weakly supervised manner. The main idea is to leverage the temporal information (*e.g.*, tracks) to extend weakly-labeled instances or potential object proposals across frames, which are then used as pseudo-labels to re-train the detectors. In the context of UDA, Tang et al. (2012); RoyChowdhury et al. (2019); Ošep et al. (2017) also incorporate object tracks to discover high quality pseudo-labels for self-training.

A key difference between these methods and ours is that we do not just interpolate tracks, but also *extrapolate* these tracks to infer objects when they are too far away to be detected accurately. We are able to do this by operating in 3D and leveraging the dynamics of objects (by physical-based motion models). In contrast, most of the methods above may disregard faraway objects (that appear too small in the image) because their tracks are unreliable (Ošep et al., 2017). Our approach also differs from this prior work in the application domains and data modalities: we focus on LiDAR-based 3D object detection, as opposed to 2D object detectors. Specifically, we exploit the property that objects in 3D are scale-invariant to correct object sizes along tracks, which is not applicable in 2D.

## 3 EXPLOITING PLAYBACKS FOR UDA

Similar to most published work on 3D object detection for autonomous driving, we focus on frame-wise 3D detectors. A detector is first trained on a source domain, and is then applied in a target domain.(*e.g.*, a new city). Wang et al. (2020) conducted a comprehensive analysis and revealed a drastic performance drop in such a scenario: many of the target objects are either misdetected or mislocalized, especially if they are far away. To aid adaptation, we assume access to an unlabeled dataset of *video sequences* in the target domain, which could simply be recordings of the vehicle's sensors while it was in operation. Our approach is to generate pseudo-labels for these recordings that can be used to adapt the detector to the new environment during periods in which the car is not in use. We do not assume access to the source data when performing adaptation—it is unlikely that the car producer will share its data with the customers after the detector is deployed.

### 3.1 TRACKING FOR IMPROVED DETECTION

One approach to improve the test accuracy based on the frame-wise detection outputs is *on-line* tracking by detection (Breitenstein et al., 2010; 2009; Hua et al., 2015). Here, detected objects are associated across current and *past* frames to derive trajectories, which are used to filter out false positives, impute false negatives, and adjust the initial detection bounding boxes in the current frame.

**On-line 3D object tracking.** We investigate this idea with a Kalman Filter based tracker (Diaz-Ruiz et al., 2019; Chiu et al., 2020; Weng et al., 2020), which has shown promising results in benchmark

tracking leader boards (Caesar et al., 2019). We opt to not use a learning-based tracker (Yin et al., 2020b) as such trackers would also require adaptation before they can be applied to improve detection in the target domain. Specifically, we apply the tracker by Diaz-Ruiz et al. (2019). The algorithm estimates the joint probability $p(\mathbf{a}_k, \mathbf{x}_k | \mathbf{z}_k)$ at time $k$, where $\mathbf{x}_k$ is the set of tracked object states (*e.g.*, cars speeds and locations), $\mathbf{z}_k$ is the set of observed sensor measurements (here each measurement is a frame-wise detection), and $\mathbf{a}_k$ the assignment of measurements to tracks. The joint distribution can be decoupled into the continuous estimation problem $p(\mathbf{x}_k | \mathbf{a}_k, \mathbf{z}_k)$, which is solved recursively via an Extended Kalman Filter (EKF), and the discrete data assignment, $p(\mathbf{a}_k | \mathbf{x}_k, \mathbf{z}_k)$, which is solved via Global Nearest-Neighbor (GNN). The EKF parameterizes the state $\mathbf{x}$ of a single ($i$th) object as a vehicle (position, velocity, and shape) *relative* to the ego-vehicle

$$\mathbf{x}_k^i = \begin{bmatrix} x & y & \theta & s & l & w \end{bmatrix}^T , \tag{1}$$

where $x$, $y$ are the location of the tracked vehicle's back axle relative to a fixed point on the ego-vehicle, $\theta$ is the vehicle orientation relative to the ego-vehicle, $s$ is the absolute ground speed, and $l, w$ are the length and width. The EKF uses a dynamic model of the evolution of the state over time. Here we assume that the tracked vehicle is moving at a constant speed and heading in the global coordinate frame, with added noise to represent the uncertainty associated with vehicle maneuvers. This tracker has been shown to work well on tracking moving objects from a self-driving car (Miller et al., 2011; Diaz-Ruiz et al., 2019). More details of the tracker are given in the supplementary. As will be seen in subsection 4.2, applying this tracker can indeed improve the detection accuracy online, via imputing missing detection, correcting mislocalized detection, and rejecting wrong detection in $\mathbf{z}_k$ at current time $k$ by $\mathbf{x}_k$.

**Off-line 3D object tracking.** Online trackers are constrained to only use past information in order to improve current detections. Relaxing this constraint for off-line tracking (*e.g.*, to be able to look into the future and come back to the current time), we can obtain even more accurate estimates of vehicle states. While such an improvement is not applicable directly during test time, higher accuracy tracking on unlabeled driving sequences will be highly valuable to adapt the source detector in a self-supervised fashion, as we will explain in the following section.

### 3.2 Self-training for UDA

Self-training is a simple yet fairly effective way to improve a model with unlabeled data (McClosky et al., 2006b;a; Kumar et al., 2020; Lee, 2013; Chen et al., 2011). The basic idea is to apply an existing model to an unlabeled data set and use the high confidence predictions (here detections), which are likely to be correct, as "pseudo-labels" for fine-tuning. One key to success for self-training is the quality of the pseudo-labels. In particular, we desire two qualities out of the detections we use as pseudo-labels. First, they should be *correct*, i.e., they should not include false positives. Second, they should have *high coverage*, i.e., they should cover all cases of objects. Choosing high confidence detections as pseudo-labels satisfies the first criterion but not the second. With 3D object detection, we find that most of the high confidence examples are easy cases: unoccluded objects near the self-driving car. This is where offline tracking becomes a crucial component to include the more challenging cases (far away, or partially occluded objects) in the pseudo-label pool.

### 3.3 High quality pseudo-labels via 3D off-line tracking

How do we obtain pseudo-labels for far-away, hard-to-detect objects that the detector cannot reliably detect? We propose to exploit tracking by leveraging two facts in the autonomous driving scenario. First, the available unlabeled data is in the form of *sequences* (akin to videos) of point clouds over time. Second, the objects of interest and the self-driving car move in fairly constrained ways. We will run the object detector on *logged* data, so that we can easily analyze both forwards and backwards in time. The object detector will detect objects accurately only when they are close to the self-driving car. Once detected over a few frames, we can estimate the object's motion either towards the car or away from it, and then both *interpolate* the object's positions in frames where it was missed, or *extrapolate* the object into frames where it is too far away for accurate detection. We show an example of this procedure in Figure 1. Through dynamic modeling, tracking, and smoothing over time we can correct noisy detections; and with extrapolation and interpolation, we can recover far away missed detections.

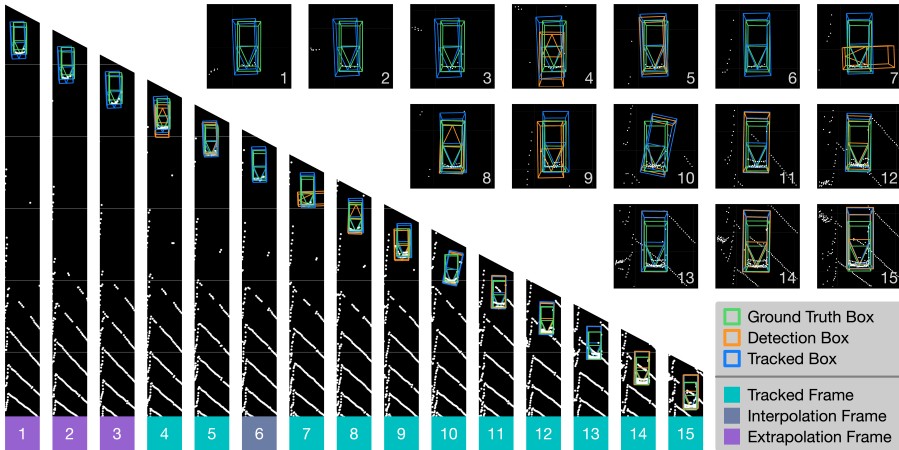

Figure 1: Example track pointcloud of a vehicle moving towards ego vehicle where pseudo-labels are recovered through extrapolation, interpolation, and smoothing. Pseudo-labels gained are instances where estimated bounding box (blue) is observed, while the original detection (orange) was missing or poorly aligned with ground truth (green). Better viewed in color. Zoom in for details.

Concretely, we proposed to augment the following functionalities that utilizes the future information into the on-line tracker introduced in subsection 3.1, turning it into an off-line tracker specifically designed to improve detection, *i.e.*, generating higher quality pseudo-labels, for self-training.

**State smoothing.** Frame-wise 3D object detectors can generate inconsistent, noisy detection across time (e.g., frame 4, 7 and 9 in Figure 1). The model-based tracking approach in subsection 3.1 reduces this noise, but we can go further by smoothing tracks back and forth over time, since our data is off-line. In this work, we use a fixed-point Rauch-Tung-Striebel (RTS) smoother (Bar-Shalom et al., 2001) to smooth the tracked state estimates. See the supplementary material for details.

**Adjusting object size.** As shown in Wang et al. (2020), the distribution of car sizes in different domains (e.g. different cities) can be different. As such, when tested on a novel domain, detectors often predict incorrect object sizes. This is especially true when the LiDAR signal is too sparse for correct size information. We can also use our tracking to correct such systematic error. Assuming that the most confident detections are more likely to be accurate, we estimate the size of the object by averaging the size of the three highest scoring detections. We use this size for all objects in this track.

**Interpolation and extrapolation.** We use estimation (forward in time) and smoothing (backward in time) to recover missed detections, and in turn, to increase the recall rate of pseudo-labels (e.g., frame 1 – 3 and 6 in Figure 1). If a detection is missed in the middle of a track, we restore it by taking the estimated state from smoothing. We also extrapolate the tracks both backward and forward in time, so that tracks that were prematurely terminated due to missing detections, can be recovered. Most commonly, we are able to recover detections of vehicles that were lost as they moved away from the ego vehicle because the measurement signals became sparser (or in turn, which started far away and were then only detected when the vehicle got close enough). Extrapolations are performed by first using dynamic model predictions of the EKF to predict potential bounding boxes; measurements are obtained by performing a search and detection in the vicinity of the prediction. We apply the detector in a 3 m$^2$ area around the extrapolated prediction, yielding several 3D bounding box candidates. After filtering out candidates with confidence lower than some threshold, we select the candidate with highest BEV IoU with the prediction as a measurement. If the track loses such measurement for three consecutive frames, extrapolations are stopped. With this targeted search, we are able to recover objects that were missed due to low confidence. After extrapolating and interpolating detections for all tracks, we perform Non Maximum Suppression (NMS) over bounding boxes in BEV, where more recent extrapolations/interpolations are prioritized.

**Discussion.** The tracker we apply is standard and simple. We opt it to show the power of our dreaming approach for UDA—exploiting off-line, forward and backward information to derive high-quality pseudo-labels for adapting detectors. More sophisticated trackers will likely improve the results further. While we focus on frame-wise 3D detectors, our algorithm can be applied to adapt video-based 3D object detectors (Yin et al., 2020a) as well.

Table 1: We summarize properties of five datasets (or subsets of them). *For nuScenes, we use 10Hz data to generate pseudo-labels, but subsample them in 2Hz (12562 frames) afterwards.

| Dataset | Location | Train Set | | | | # Test Frame |
|---------|----------|-----|-------|---------------|-----------------|--------------|
| | | FPS | # Seq | # Total Frame | # Labeled Frame | |
| KITTI | Karlsruhe | 10 Hz | 96 | 13596 | 3712 | 3769 |
| Argo | Miami and Pittsburgh | 10 Hz | 65 | 13122 | 13122 | 5014 |
| Lyft | Palo Auto | 5 Hz | 100 | 12599 | 12599 | 3011 |
| nuScenes | Boston | 10 Hz | 312 | 61121* | 12562 | 3133 |
| Waymo | San Francisco | 10 Hz | 60 | 11886 | 11886 | 19828 |

One particular advantage of fine-tuning on the pseudo-labeled target data is to allow the detector not only adapting its predictions (*e.g.*, the box regression) but also its features (*e.g.*, the early layers in the neural networks) to the target domain. The resulting detector therefore can usually lead to more accurate detection than the pseudo-labels it has been trained on.

## 4 EXPERIMENTS

**Datasets.** We experiment with five autonomous driving data sets: KITTI (Geiger et al., 2012; 2013), Argoverse (Chang et al., 2019), nuScenes (Caesar et al., 2019), Lyft (Kesten et al., 2019) and Waymo (Sun et al., 2019). All datasets provide LiDAR data in sequences and ground-truth bounding box labels for either all or part of the data. We briefly summarize these five datasets in Table 1. We follow a setup similar to that in Wang et al. (2020), but with different splits on some of the datasets in order to keep the training and test sets non-overlapping with sequences.

**UDA settings.** We train models from the source domain using labeled frames. *We use the train set (and its pseudo-labels) of the target domain to adapt the source detector, and evaluate the adapted detector on the test set.* See supplementary for more details.

**Metric.** We follow KITTI to evaluate object detection in 3D and the BEV metrics. As it has been the main focus of existing works on 3D object detection, we focus on the *Car* category. We report average precision (AP) with the intersection over union (IoU) thresholds at 0.5 or 0.7, *i.e.*a car is correctly detected if the IoU between it and the predicted box is larger than 0.5 or 0.7. We denote AP for the 3D and BEV by $AP_{3D}$ and $AP_{BEV}$, respectively. Because on the other datasets there is no official separation on the difficulty levels like in KITTI, we split AP by depth range.

**3D object detection models.** We use two LiDAR-based models POINTRCNN (Shi et al., 2019) and PIXOR (Yang et al., 2018) to detect objects in 3D. They represent two different but popular ways of processing point cloud data. POINTRCNN uses PointNet++ (Qi et al., 2017b) to extract point-wise features, while PIXOR applies 2D convolutions in BEV of voxelized point clouds. Neither relies on images. We mainly report and discuss results of POINTRCNN.

**Hyper-parameters.** To train detectors on source domain, we use the hyper-parameters provided by Shi et al. (2019) for POINTRCNN; for PIXOR, we follow Wang et al. (2020) to train it using RMSProp with momentum 0.9 and learning rate $5 \times 10^{-5}$ (decreased by a factor of 10 after 50 and 80 epochs) for 90 epochs. For self-training on the target domain, we initialize from the pre-trained model on the source domain. For POINTRCNN, we fine-tune the model with learning rate $2 \times 10^{-4}$ and 40 epochs in RPN and 10 epochs in RCNN. For PIXOR, we use RMSProp with momentum 0.9 and learning rate $5 \times 10^6$ (decreased by a factor of 10 after 10 and 20 epochs) for 30 epochs.

We developed and tuned our dreaming method with Argoverse as the source domain and KITTI as the target domain (in the target domain, we only use the training set). We then fixed all hyper-parameters for all subsequent experiments. We list other relevant hyper-parameters in the supplementary.

### 4.1 BASELINES

We compare against two baselines under the UDA setting.

**Self-Training (ST).** We apply a self-training scheme similar to that typically used in the 2D problems (Kumar et al., 2020). When adapting the model from the source to the target, we apply the source model to the target training set. We then keep the detected cars of confidence scores $> 0.8$

Table 2: **Unsupervised domain adaptation from Argoverse to KITTI.** We report $AP_{BEV}$/ $AP_{3D}$ of the **car** category at IoU $= 0.7$ and IoU $= 0.5$ across different depth range, using POINTRCNN model. ST stands for Self-Training (Kumar et al., 2020), SN stands for Statistical Normalization (Wang et al., 2020). Our method *Dreamt* is marked in blue. We show the performance of in-domain model, *i.e.*, the model trained and evaluated on KITTI, at the first row in gray. We also show results by directly applying online and offline (no feasible in real-time) tracking. Best viewed in color.

| Range(m) | IoU 0.5 | | | IoU 0.7 | | |
|---|---|---|---|---|---|---|
| Method | 0-30 | 30-50 | 50-80 | 0-30 | 30-50 | 50-80 |
| in-domain | 90.0 / 89.9 | 81.0 / 79.9 | 40.4 / 36.3 | 89.0 / 78.0 | 70.3 / 51.5 | 26.6 / 9.8 |
| no retraining | 89.4 / 88.8 | 71.0 / 65.2 | 18.0 / 13.9 | 72.6 / 47.8 | 35.8 / 14.6 | 4.9 / 3.0 |
| no retraining + online | 89.3 / 88.5 | 71.9 / 66.1 | 18.3 / 14.0 | 72.3 / 47.5 | 38.2 / 14.9 | 5.5 / 1.2 |
| no retraining + offline | 89.3 / 88.8 | 72.7 / 67.7 | 18.9 / 15.0 | 74.5 / 49.6 | 43.5 / 18.1 | 7.3 / 1.8 |
| ST | **89.5 / 89.2** | 73.2 / 68.5 | 28.2 / 22.5 | 76.8 / 52.9 | 44.2 / 20.6 | 13.1 / 2.1 |
| Dreaming | 89.3 / **89.2** | **74.6 / 72.4** | **30.1 / 25.6** | **77.6 / 54.7** | **49.9 / 24.3** | **14.5 / 3.4** |
| SN only | 89.3 / 88.2 | 69.6 / 65.4 | 14.6 / 13.3 | 83.8 / 59.1 | 53.5 / 27.2 | 9.3 / 3.6 |
| SN only + online | 89.4 / 88.2 | 68.8 / 65.4 | 19.4 / 16.2 | 83.5 / 60.1 | 50.7 / 27.0 | 13.4 / 9.5 |
| SN only + offline | 89.4 / 88.2 | 68.9 / 66.1 | 21.8 / 19.3 | 83.3 / 59.2 | 50.9 / 26.9 | 13.8 / 9.8 |
| SN + ST | **89.8** / 89.3 | 74.4 / 72.8 | 22.3 / 21.3 | **87.0** / 70.4 | 62.2 / 37.7 | 16.4 / 7.1 |
| SN + Dreaming | **89.8 / 89.4** | **75.4 / 73.8** | **29.4 / 25.4** | **87.0 / 73.5** | **62.8 / 41.9** | **17.2 / 10.3** |

Table 3: **Dreaming results on unsupervised adaptation among five auto-driving datasets.** Here we report $AP_{BEV}$ and $AP_{3D}$ of the Car category on range 50-80m at IoU$= 0.5$. On each entry (*row*, *column*), we report AP of UDA from *row* to *column* in the order of no fine-tuning / ST / Dreaming. At the diagonal entries, we report the AP of in-domain model. Our method is marked in blue.

| $AP_{BEV}$ | KITTI | Argoverse | Lyft | nuScenes | Waymo |
|---|---|---|---|---|---|
| KITTI | 40.4 | 20.1 / 26.9 / **28.4** | 49.6 / 56.3 / **56.4** | 1.4 / **9.1** / 4.5 | 42.8 / 48.5 / **50.2** |
| Argoverse | 18.0 / 28.2 / **30.1** | 37.8 | 46.3 / 48.8 / **54.5** | 0.6 / **9.1** / 3.0 | 50.4 / 50.9 / **56.0** |
| Lyft | 26.1 / 30.8 / **33.9** | 29.3 / 30.7 / **35.2** | 67.2 | 4.5 / **9.1** / 6.0 | 51.6 / 51.9 / **56.9** |
| nuScenes | 9.6 / 16.4 / **21.7** | 3.0 / 12.4 / **17.7** | 22.9 / 39.1 / **46.4** | 3.5 | 24.9 / 42.5 / **50.1** |
| Waymo | 14.3 / 25.6 / **27.8** | 24.7 / 23.4 / **28.3** | 45.3 / 54.0 / **55.5** | 0.2 / 3.0 / **9.1** | 58.2 |
| $AP_{3D}$ | KITTI | Argoverse | Lyft | nuScenes | Waymo |
| KITTI | 36.3 | 15.4 / 19.5 / **20.8** | 40.0 / 46.3 / **47.7** | **0.9** / 0.4 / 0.4 | 33.4 / 39.3 / **41.0** |
| Argoverse | 13.9 / 22.5 / **25.6** | 30.0 | 42.9 / 46.0 / **47.6** | 0.1 / 0.2 / **3.0** | 47.8 / 48.2 / **49.1** |
| Lyft | 20.4 / 24.0 / **26.4** | 25.3 / 26.7 / **27.3** | 65.5 | 0.4 / **9.1** / 1.1 | 49.6 / 50.6 / **50.8** |
| nuScenes | 5.5 / 8.7 / **13.4** | 3.0 / 9.1 / **13.0** | 15.1 / 30.2 / **37.8** | 2.8 | 22.5 / 39.6 / **45.5** |
| Waymo | 8.5 / 18.1 / **19.7** | 19.6 / 21.3 / **22.3** | 43.0 / 48.1 / **49.6** | 0.0 / 0.3 / **9.1** | 50.8 |

(label-sharpening) and use them as pseudo-labels to fine-tune the model. We select the threshold following our hyper-parameter selection procedure and apply it to all the experiments.

**Statistical Normalization (SN).** Wang et al. (2020) showed that car sizes vary between domains: popular cars at different areas can be different. When the mean bounding box size in the target domain is accessible, either from limited amount of labeled data or statistical data, we can apply *statistical normalization (SN)* (Wang et al., 2020) to mitigate such a systematic difference in car sizes. SN adjusts the bounding box sizes and corresponding point clouds in the source domain to match those in the target domain, and fine-tunes the model on such "normalized" source data, with no need to access target sensor data. We follow the exact setting in (Wang et al., 2020) to apply SN.

## 4.2 EMPIRICAL RESULTS

**Adaptation from Argoverse to KITTI.** We compare the UDA methods under Argoverse to KITTI in Table 2 and observe several trends: 1) Models experience a smaller domain gap when objects are closer (0-30 m vs 30-80 m); 2) Though directly applying on-line tracking can improve the detection performance, models improve more after just self-training; 3) The off-line tracking is used to provide extra pesudo-labels for fine-tuning, and interestingly, models fine-tuned from pseudo-labels can out-perform pseudo-labels themselves; 4) DREAMING improves over ST and SN by a large margin, especially on IoU at 0.5; 5) DREAMING introduces a large gain in AP for faraway objects, *e.g.*, on range 50-80 m, compared with ST, it boosts the $AP_{BEV}$ on IoU at 0.5 from 28.2 to 30.1.

Table 4: **UDA from KITTI (city, campus) to KITTI (road, residential)**. Naming is as in Table 2.

| Range(m) Method | IoU 0.5 | | | IoU 0.7 | | |
|---|---|---|---|---|---|---|
| | 0-30 | 30-50 | 50-80 | 0-30 | 30-50 | 50-80 |
| no retraining | 89.4 / 89.4 | 79.7 / 77.9 | 36.2 / 30.9 | 88.3 / 77.9 | 66.0 / 47.5 | 19.2 / 6.2 |
| ST | 89.4 / 89.4 | 78.0 / 76.7 | 35.9 / 31.2 | 88.2 / 77.8 | 66.3 / 48.3 | 19.0 / 7.7 |
| Dreaming | **89.6 / 89.6** | **80.1 / 78.9** | **41.4 / 35.3** | **88.5 / 78.2** | **68.0 / 49.6** | **23.2 / 12.7** |

Table 5: **Ablation study of UDA from Argoverse to KITTI.** We report $AP_{BEV}$/ $AP_{3D}$ of the **car** category at IoU = 0.5 and IoU = 0.7 across different depth range, using POINTRCNN model. Naming is as in Table 2. S stands for smoothing, R for resizing, I for interpolation and E for extrapolation. The last row is our full approach.

| Range(m) Method | IoU 0.5 | | | IoU 0.7 | | |
|---|---|---|---|---|---|---|
| | 0-30 | 30-50 | 50-80 | 0-30 | 30-50 | 50-80 |
| no retraining | 89.4 / 88.8 | 71.0 / 65.2 | 18.0 / 13.9 | 72.6 / 47.8 | 35.8 / 14.6 | 4.9 / 3.0 |
| ST | 89.5 / 89.2 | 73.2 / 68.5 | 28.2 / 22.5 | 76.8 / 52.9 | 44.2 / 20.6 | 13.1 / 2.1 |
| ST + S | 89.5 / 89.3 | 73.7 / 68.7 | 28.3 / 22.3 | 76.8 / 53.4 | 45.2 / 21.5 | 12.9 / **4.7** |
| ST + S + R | **89.5 / 89.3** | 74.0 / 71.6 | 28.3 / 23.9 | **78.0 / 55.2** | **50.8** / 23.9 | 10.3 / 2.7 |
| ST + S + R + I | 89.3 / 89.1 | 73.9 / 71.6 | 28.1 / 23.3 | 77.6 / 54.5 | 50.4 / 24.0 | 11.2 / 3.4 |
| ST + S + R + I + E | 89.4 / 89.2 | **74.9 / 72.5** | **31.0 / 25.7** | 77.8 / 55.1 | 50.4 / **24.1** | **14.3** / 3.3 |

**Adaptation among five datasets.** We further applied our methods to adaptation tasks among the five datasets. Due to limited space, we show the results of $AP_{BEV}$ and $AP_{3D}$ on range 50-80 m at IoU = 0.5 in Table 3 (see supplementary material for results at IoU = 0.7 on other ranges and results with SN). Our method consistently improves the adaptation performance on faraway ranges, while having mostly equal or better performance on close-by ranges.

**Adaptation between different locations inside the same dataset.** Different datasets not only come from different locations but also use different sensor configurations. To isolate the effects of the former (which is our motivating application), in Table 4 we evaluate our method's performance for domain adaptation within the KITTI dataset from city/campus scenes to residential/country scenes (details in supplementary). Our method consistently outperforms no fine-tuning and ST, especially on 30-50 m and 50-80 m range.

**Ablation Study.** We show ablation results in Table 5. Here we fine-tune models using ST and adding smoothing (S), resizing (R), interpolation (I) and extrapolation (E) to the pseudo-label generation. It can be observed that ST alone already boosts performance considerably. Through selecting high confidence detections, smoothing and adjusting the object size we ensure that the pseudo-labels provided are mostly *correct*. But just these do not address the second criteria for desired pseudo-labels: *high coverage*. We observe noticeable boosts when interpolation and extrapolations are added, specially for far away objects. This is due to extrapolations and interpolations recovering pseudo-labels for low confidence or missed detections for distant vehicles.

**Others.** We show more results, including adaptation results on PIXOR detectors, qualitative visualizations and analysis on pseudo-labels, in the supplementary material.

## 5 CONCLUSION AND DISCUSSION

In this paper, we have introduced a novel method towards closing the gap between source and target in unsupervised domain adaptation for LiDAR-based 3D object detection. Our approach is based on self-training, while leveraging vehicle dynamics and offline analysis to generate pseudo-labels. Importantly, we can generate high quality pseudo-labels even for difficult cases (i.e. far away objects), which the detector tends to miss before adaptation. Fine-tuning on these pseudo-labels improves detection performance drastically in the target domain. It is hard to conceive an autonomous vehicle manufacturer that could collect, label, and update data for every consumer environment, meeting the requirements to allow self-driving cars to operate everywhere freely and safely. By significantly reducing the adaptation gap between domains, our approach takes a significant step towards making this vision a reality nevertheless.

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

# SUPPLEMENTARY MATERIAL

We provide details omitted in the main text.

- Appendix A: additional related work (cf. section 2 of the main paper).
- Appendix B: additional details on 3D object tracking (cf. subsection 3.3 of the main paper).
- Appendix C: additional details on hyper-parameters (cf. section 4 of the main paper).
- Appendix D: additional details on the datasets (cf. section 4 and subsection 4.2 of the main paper).
- Appendix E: additional analysis (cf. subsection 4.2 of the main paper).
- Appendix F: adaptation results using PIXOR detector
- Appendix G: qualitative comparison on baselines and our method (cf. subsection 4.2 of the main paper).
- Appendix H: additional results on unsupervised domain adaptation among five autonomous driving datasets (cf. subsection 4.2 of the main paper).

## A   RELATED WORK

Unsupervised domain adaptation (UDA) has recently be extended to driving scenes, but mainly for 2D semantic segmentation (Chen et al., 2018b; Hoffman et al., 2018; Huang et al., 2018; Luo et al., 2019; Saito et al., 2018; Saleh et al., 2018; Sankaranarayanan et al., 2018; Tsai et al., 2018; Zhang et al., 2017; Zou et al., 2018; Chen et al., 2017) and 2D object detection (Cai et al., 2019; Chen et al., 2018a; He & Zhang, 2019; Hsu et al., 2020; Khodabandeh et al., 2019; Kim et al., 2019b; Rodriguez & Mikolajczyk, 2019; Saito et al., 2019; Wang et al., 2019a; Zhuang et al., 2020; Zhu et al., 2019).

## B   3D OBJECT TRACKING

**Baseline on-line tracking.**   Given a set of detections (in this case the 3D bounding box detections), and the predicted states of all tracks $\hat{\mathbf{x}}_{k+1}$, we use a GNN to estimate the detection to track association $p(\mathbf{a}_{k+1}|\mathbf{x}_{k+1}, \mathbf{z}_{k+1})$ at each timestep. This is formulated as a linear assignment problem, where the cost to be minimized is the bird's eye view (BEV) IoU between the predicted box and the measurement. We initialize track when $c_{\min-\mathrm{hits}}$ measurements to the same track are realized. We end a track when it does not obtain any measurement updates over $c_{\max-\mathrm{age}}$ frames, or the object vehicle exits the field of view (FOV).

Given the measurement assignment, the EKF updates the state distribution $p(\mathbf{x}_{k+1}|\mathbf{a}_{k+1}, \mathbf{z}_{k+1})$ with the measurement detection $\mathbf{z}_{k+1}$. The measurement likelihood is computed via an uncertainty model derived in Diaz-Ruiz et al. (2019). With the dynamics model, associations and measurement models, the EKF predicts and updates state distributions in the form of a state estimate and error covariance, $\hat{\mathbf{x}}_{k|k}, P_{k|k}$.

**State smoothing.**   Smoothing requires a backward iteration ($k = N, N-1, ...1$) that is performed after the forward filtering where, $\bar{\mathbf{x}}_{k|k}\ P_{k|k}$, the a-posteriori state and state error covariance estimates, and $\bar{\mathbf{x}}_{k+1|k}\ P_{k+1|k}$ , the a-priori state and state error covariance estimates have been calculated. The smother gain, $C_k$, is obtained from

$$C_k = P_{k|k}F_k^T P_{k+1|k}^{-1} \tag{2}$$

where $F_k$ is the Jacobian of the dynamics model evaluated at $\bar{\mathbf{x}}_{k|k}$. The smoothed state is then evaluated as

$$\bar{\mathbf{x}}_{k|N} = \bar{\mathbf{x}}_{k|k} + C_k[\bar{\mathbf{x}}_{k+1|N} - \bar{\mathbf{x}}_{k|k}] \tag{3}$$

while the covariance of the smoothed state is evaluated as

$$P_{k|N} = P_{k|k} + C_k[P_{k+1|N} - P_{k+1|k}]C_k^T \tag{4}$$

## C    HYPER-PARAMETERS

In performing the forward pass of the Extended Kalman Filter (EKF) to calculate the a-posteriori state and state error covariance estimates (*i.e.*, $\bar{\mathbf{x}}_{k|k}$ $P_{k|k}$) and the a-priori state and state error covariance (*i.e.*, $\bar{\mathbf{x}}_{k+1|k}$ $P_{k+1|k}$) in subsection 3.3, a process noise covariance matrix ($Q$), measurement noise covariance matrix ($R$), initial state estimate ($\bar{\mathbf{x}}_0$), and initial state error covariance matrix ($P_0$) are necessary. The measurement noise matrix is established through measurement variance, and was obtained by observing the variance of position ($x, y$), orientation ($\theta$), length ($l$) and width ($w$) errors from testing detector in Argoverse to KITTI setting. A larger measurement noise matrix is used in the EKF for extrapolation as larger variance was observed with far-away detections. The process noise matrix should represent the magnitudes of dynamical noise that the system might experience. In the model in Diaz-Ruiz et al. (2019), which is constant velocity and heading, there are seven noise parameters: $e_\theta, e_s, e_{v_x}, e_{v_y}, e_{\omega_z}, e_l, e_w$ which correspond to the diagonal of the $Q$ matrix. The variables $e_\theta, e_s, e_l, e_w$ are modeled as zero mean, mutually uncorrelated, Gaussian, and white process noise. Intuitively, $e_\theta, e_s$ represent the uncertainty associated with the orientation and speed of vehicles, especially given that the model assumes constant speed straight line motion. The noise associated with the orientation and speed of the target vehicles are the largest and of most importance. The time derivative of the objects length and width are zero, where $e_l, e_w$ are small tuning parameters which control the response of the filter. While $e_{v_x}, e_{v_y}, e_{\omega_z}$ are noises associated with the pose of ego-vehicle and small values were chosen given that a high accuracy in ego-vehicle pose is expected across the datasets. Finally, the EKF necessitates an initialization for the state and state error covariance estimates. The initial detection was used for state initialization, and relatively large conservative uncertainties were used for state error covariance.

- Extended Kalman Filter (EKF) and Global Nearest Neighbor (GNN) parameters for tracking and data association (cf. subsection 3.3):

    1) Measurement noise covariance matrix:
    $$R = \text{diag}(0.1\text{m}^2, 0.1\text{m}^2, 0.015\text{rad}^2, 0.07\text{m}^2, 0.04\text{m}^2)$$
    2) Process noise covariance matrix:
    $$Q = \text{diag}(0.1218\tfrac{\text{rad}}{\text{s}}^2, 1\tfrac{\text{m}}{\text{s}^2}^2, 0.00545\tfrac{\text{rad}}{\text{s}}^2, 0.00545\tfrac{\text{m}}{\text{s}}^2, 0.00307\tfrac{\text{rad}}{\text{s}}^2, 0.01\tfrac{\text{m}}{\text{s}}^2, 0.01\tfrac{\text{m}}{\text{s}}^2)$$
    3) State error covariance matrix initialization:
    $$P_0 = \text{diag}(2\text{m}^2, 2\text{m}^2, 0.1\text{rad}^2, 5\tfrac{\text{m}}{\text{s}}^2, 0.5\text{m}^2, 0.32\text{m}^2)$$
    4) The initial state estimate, $\bar{\mathbf{x}}_0$, is set to be the first detection values.
    5) The data association threshold (in **BEV IoU**) in GNN is set to be 0.3.
    6) The fraction of distance between the vehicle center and back-axle is $\frac{1}{4}l$, as in Diaz-Ruiz et al. (2019).
- EKF parameters for extrapolation:

    1) Measurement noise covariance matrix:
    $$R = \text{diag}(0.5\text{m}^2, 0.5\text{m}^2, 0.06\text{rad}^2, 0.07\text{m}^2, 0.04\text{m}^2)$$

We set $c_{\text{min−hits}} = 3$ and $c_{\text{max−age}} = 3$. When doing extrapolation, we use $-25$ and $-3$ as thresholds for POINTRCNN and PIXOR models respectively.

## D    DETAILS ON DATASETS

As summarized in Table 1 of the main paper, we split each dataset into two parts, a training set and a test set. When a dataset is used as the source, we train the detector using its (ground truth) training labeled frames. When a dataset is used as the target, we adapt the detector using its training sequences without revealing the ground truth labels. We evaluate the adapted models on the test set. We provide detailed properties of the five autonomous driving datasets and the way we split the data as follows.

**KITTI.** The KITTI object detection benchmark (Geiger et al., 2013; 2012) contains 7,481 scenes for training and 7,518 scenes for testing. All the scenes are pictured around Karlsruhe, Germany in clear weather and day time. For each scene, KITTI provides 64-beam Velodyne LiDAR point cloud and stereo images. The training set is further separated into 3,712 training and 3,769 validation scenes as suggested by Chen et al. (2015). The training scenes are sampled from 96 data sequences, which

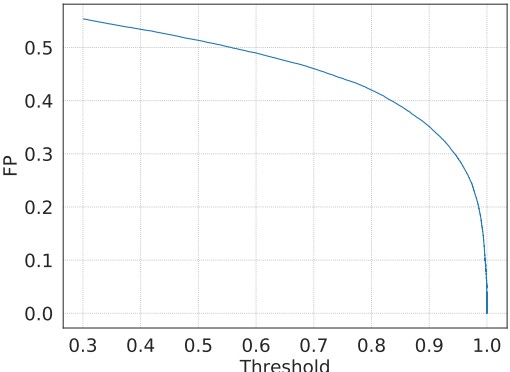

Figure 2: False positive rates above detection confidence for a model trained on Argoverse and tested on KITTI.

have no overlap with the sequences where validation scenes are sampled. These training sequences are collected in 10 Hz, resulting in 13,596 frames. We extract the sequences from the raw KITTI data as our adaptation data. We use such data splits in all experiments related to KITTI, except for **adaptation between different locations inside the same dataset** (cf. Table 4 of the main paper). For adaptation between different locations inside the KITTI dataset, we split the sequences in training set by their categories: *city/campus* as the source (38 sequences, 9,556 frames, 3,420 labeled frames) and *residential/road* as the target (23 sequences, 10,448 frames, 3619 labeled frames). In Table 4, all models are pre-trained in city/campus data. Due to limited data, different from what we do UDA among five autonomous driving datasets, *we perform adaptation and final evaluation on the full target data.* Note that in this case, the detectors still do not have access to ground truth labels during UDA.

**Argoverse.** The Argoverse dataset (Chang et al., 2019) is collected around Miami and Pittsburgh, USA in multiple weathers and during different times of the day. For each scene (timestamp), Argoverse provides a 64-beam LiDAR point cloud captured by stacking two 32-beam Velodyne LiDAR vertically. We extracted synchronized images (from front camera) and corresponding point clouds from the original Argoverse dataset. We follow the official split on training and validation sets, which contain 13,122 scenes and 5,014 scenes respectively. We use sequences in the training set (without using the ground truth labels) as our adaptation data.

**Lyft.** The Lyft Level 5 dataset collects 18,634 scenes around Palo Auto, USA in clear weather and during day time. For each scene, Lyft provides the ground-truth bounding box labels and point cloud captured by a 40 (or 64)-beam roof LiDAR and two 40-beam bumper LiDAR sensors. We follow Wang et al. (2020) and separate the dataset by sequences, resulting in 12,599 frames for training (100 sequences), 3,024 validation frames (24 sequences) and 3,011 frames for testing (24 sequences). The sequences in 5 Hz and we use training sequences without labels as adaptation data.

**nuScenes.** The nuScenes dataset (Caesar et al., 2019) collects scenes around Boston, USA and Singapore in multiple weather conditions and during different times of the day. For each scene, nuScenes provides a point cloud captured by a 32-beam roof LiDAR. We use the data collected in Boston, and sampled 312 sequences for training and 78 sequences for validation. We use 10 Hz sensor data without labels as our adaptation data (61,121 frames in total), and evaluate the model on 2 Hz labeled data in validation set (3,133 frames). Note that after generating pseudo-labels, we sub-sample adaptation data using 2 Hz into 12,562 frames.

**Waymo.** The Waymo open dataset (Sun et al., 2019) is mostly collected around San Francisco, Phoenix, and Mountain View in multiple weather conditions and at multiple times of the day. It provides point clouds captured in 10 Hz by five LiDAR sensors (one on roof, four on side) and images from five cameras. We randomly sample 60 tracks (11,886 frames) captured in San Francisco as our adaptation data, and another 100 tracks (19,828 frames) in San Francisco as our validation data.

## E    FURTHER ANALYSIS

**Analysis on Pseudo-Labels.** Figure 2 shows the (accumulated) false positive (FP) rates above a certain detector confidence threshold (setting: Argoverse → KITTI). We view a detection as a false positive if there are no ground-truth labels having an BEV IoU > 0.7 with it. We see that FP rates smoothly decrease with the increasing detector confidence, suggesting that higher confidence detection are more reliable.

**Multi-round fine-tuning.** In theory, the process of pseudo-label generation followed by fine-tuning the model can iterate for multiple rounds, yet we see a very small gain after the first round (which is what we report in all tables). We believe a curated training procedure with carefully chosen hyper-parameters is needed to make it work.

## F    ADAPTATION RESULTS USING PIXOR

We further apply our approach to the PIXOR model from Argoverse to KITTI in Table 6. Dreaming improves the accuracy at farther ranges (30-80 m) while maintaining the accuracy at close range (0-30 m). Interestingly, at IoU 0.5 in the 30-80 m ranges, we are able to surpass the in-domain performance, which uses models trained only in the target domain with the ground-truth labels. This results showcases the power of unsupervised domain adaptation (UDA): with a suitably designed algorithm, UDA that leverages both the source and target domain could outperform models trained only in a single domain.

Table 6: **Unsupervised domain adaptation from Argoverse to KITTI using PIXOR.** We report $AP_{BEV}$ of the **car** category at IoU = 0.5 and IoU = 0.7. Naming is as in Table 2.

| Range(m) | IoU 0.5 | | | IoU 0.7 | | |
|---|---|---|---|---|---|---|
| Method | 0-30 | 30-50 | 50-80 | 0-30 | 30-50 | 50-80 |
| in-domain | 88.7 | 62.6 | 21.4 | 79.6 | 49.9 | 10.0 |
| no retraining | 85.7 | 57.2 | 12.9 | 54.2 | 23.6 | 4.7 |
| ST | 86.0 | 56.3 | 12.2 | 55.3 | 24.6 | 2.5 |
| Dreaming | **87.1** | **61.1** | **20.2** | **58.0** | **28.1** | **4.8** |
| SN only | 86.7 | 58.7 | 15.1 | 76.2 | 38.7 | 5.1 |
| SN + ST | **87.4** | 58.9 | 12.4 | **78.0** | 42.4 | 3.8 |
| SN + Dreaming | **87.4** | **64.2** | **22.2** | 77.9 | **42.5** | **4.5** |

## G    QUALITATIVE RESULTS

In Figure 3, Figure 4 and Figure 5, we compare qualitatively the detection results from models trained with different adaptation strategies. We select (Argoverse, KITTI), (nuScenes, KITTI) and (nuScenes, Argoverse) as (*source*, *target*) example pairs in qualitative visualization. It can be seen that models without retraining tend to miss faraway objects. Models with self-training are able to detect some of these objects. Models with dreaming can detect more faraway objects. Self-training and dreaming both exhibit some more false positive detection.

## H    ADAPTATION AMONG FIVE DATASETS

In Table 7, we present the adaptation evaluation results across on range $0 - 80$m at IoU= 0.5 as that in Table 3. In Table 8, we present UDA results on all possible (source, target) pairs among the five autonomous driving datasets (20 pairs in total). On each pair, we show results with and without statistical normalization (Wang et al., 2020). As in Table 2, we report $AP_{BEV}$/ $AP_{3D}$ of the **car** category at IoU = 0.7 and IoU = 0.5 across different depth range, using the POINTRCNN detector. It can be seen that under these 20 UDA scenarios, our method consistently improves the adaptation performance on faraway ranges, while having mostly equal or better performance on close-by ranges.

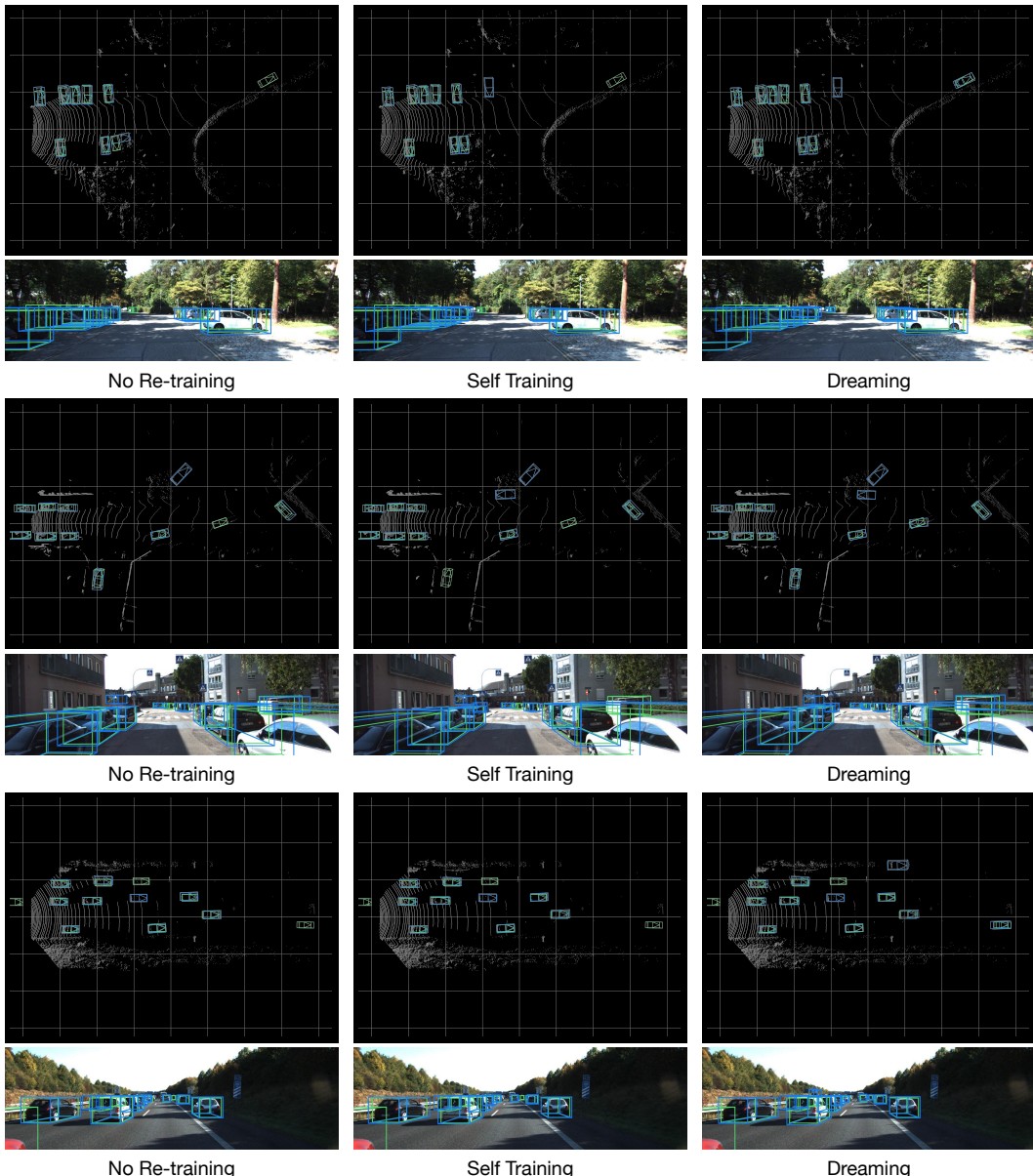

Figure 3: **Qualitative Results.** We compare the detection results on several scenes from the KITTI validation set by the POINTRCNN detectors that are trained on 1) Argoverse dataset (No Re-training), 2) Argoverse dataset and fine-tuned using self-training on KITTI (Self Training), and 3) Argoverse dataset and fine-tuned using dreaming on KITTI (Dreaming). We visualize them from both frontal-view images and bird's-eye view point maps. Ground-truth boxes are in green and detected bounding boxes are in blue. The ego vehicle is on the left side of the BEV map and looking to the right. One floor square is 10m×10m. Best viewed in color. Zoom in for details.

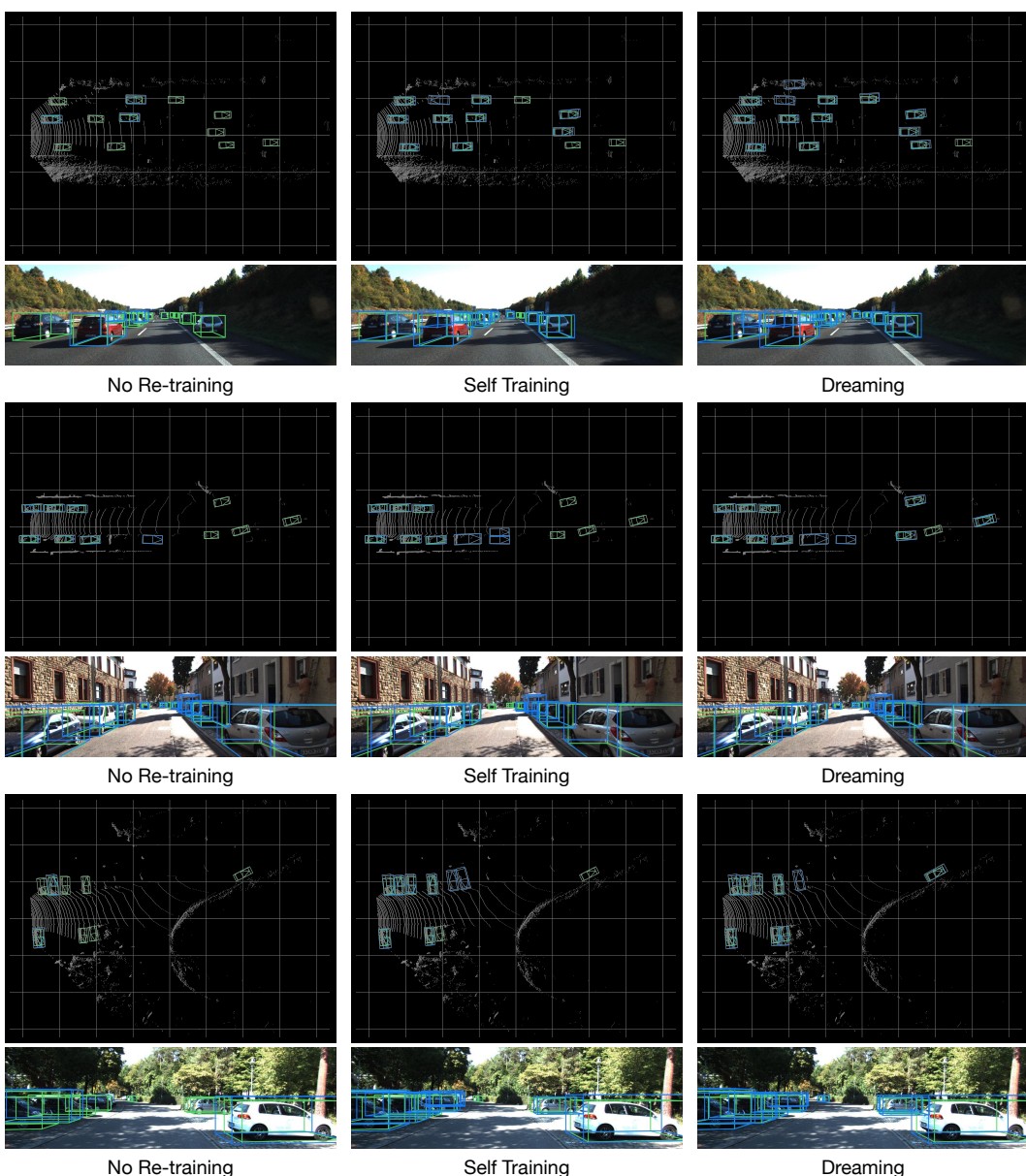

Figure 4: **Qualitative Results.** The setups are the same as those in Figure 3, but the models are pre-trained in nuScenes dataset and tested on KITTI dataset.

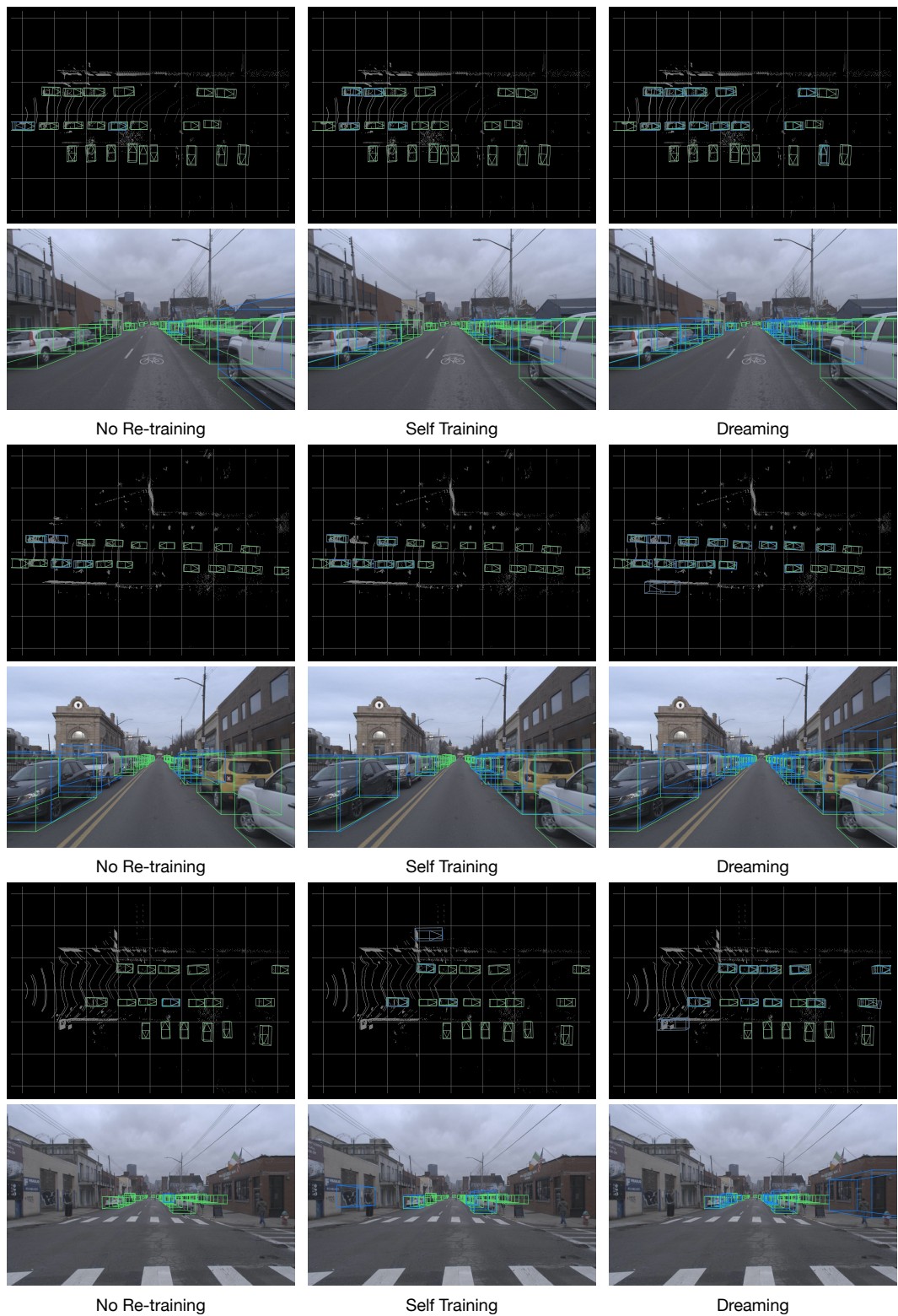

Figure 5: **Qualitative Results.** The setups are the same as those in Figure 3, but the models are pre-trained in nuScenes dataset and tested on Argoverse dataset.

Table 7: **Dreaming results on unsupervised adaptation among five auto-driving datasets.** Here we report $AP_{BEV}$ and $AP_{3D}$ of the Car category on range 0-80m at IoU= 0.5. On each entry (*row*, *column*), we report AP of UDA from *row* to *column* in the order of no re-training / ST / Dreamt. At the diagonal entries, we report the AP of in-domain model. Our method is marked in blue.

| $AP_{BEV}$ | KITTI | Argoverse | Lyft | nuScenes | Waymo |
|---|---|---|---|---|---|
| KITTI | 87.1 | 56.8 / 58.8 / **59.4** | 68.2 / 68.3 / **68.6** | 27.7 / 27.8 / **28.9** | 62.5 / 68.8 / **69.0** |
| Argoverse | 82.3 / 83.2 / **83.3** | 68.5 | 66.7 / 67.0 / **67.6** | **26.9** / 25.2 / 25.4 | 69.8 / 69.6 / **70.0** |
| Lyft | 82.6 / 84.9 / **85.3** | 60.2 / 65.7 / **66.0** | 79.2 | 28.8 / 28.2 / **29.1** | 70.6 / 70.9 / **71.1** |
| nuScenes | 61.7 / 76.5 / **79.5** | 22.4 / 38.0 / **46.6** | 41.1 / 59.0 / **65.5** | 37.7 | 51.5 / 62.2 / **68.7** |
| Waymo | 81.2 / 82.2 / **83.1** | 56.9 / 58.3 / **59.2** | 67.3 / 68.9 / **69.4** | 23.1 / 27.2 / **29.1** | 71.9 |

Table 8: **Unsupervised domain adaptation among five autonomous driving datasets.** Naming is as that in Table 2 of the main paper.

(a) KITTI to Argoverse

| Range(m) | IoU 0.5 | | | IoU 0.7 | | |
|---|---|---|---|---|---|---|
| Method | 0-30 | 30-50 | 50-80 | 0-30 | 30-50 | 50-80 |
| in-domain | 88.6 / 85.7 | 68.5 / 66.4 | 37.8 / 30.0 | 76.5 / 53.2 | 56.6 / 30.4 | 20.2 / 10.1 |
| no retraining | 79.4 / 76.5 | 51.9 / 44.6 | 20.1 / 15.4 | 53.1 / 32.9 | 29.3 / 12.3 | 5.9 / 3.0 |
| ST | 85.2 / 77.6 | 56.2 / 52.2 | 26.9 / 19.5 | 61.3 / **38.8** | 34.9 / 17.4 | 10.9 / **9.1** |
| Dreaming | **85.6 / 77.8** | **61.3 / 54.1** | **28.4 / 20.8** | **63.3** / 38.6 | **41.6 / 20.2** | **12.4** / 4.5 |
| SN only | 79.3 / 77.6 | 54.7 / 52.0 | 27.5 / 21.2 | 66.2 / 48.5 | 43.8 / 21.2 | 16.7 / 9.1 |
| SN + ST | 84.9 / 78.0 | 56.7 / 54.7 | 29.0 / 22.5 | 73.0 / **50.8** | 46.4 / **22.3** | 17.6 / 9.1 |
| SN + Dreaming | **85.1 / 78.2** | **62.3 / 55.7** | **33.6 / 26.5** | **73.1** / 50.3 | **50.6** / 20.6 | **18.5 / 9.9** |

(b) KITTI to Lyft

| Range(m) | IoU 0.5 | | | IoU 0.7 | | |
|---|---|---|---|---|---|---|
| Method | 0-30 | 30-50 | 50-80 | 0-30 | 30-50 | 50-80 |
| in-domain | 89.2 / 88.9 | 78.4 / 77.9 | 67.2 / 65.5 | 88.4 / 79.0 | 74.8 / 54.6 | 54.7 / 27.3 |
| no retraining | 81.0 / **80.8** | 73.6 / 66.8 | 49.6 / 40.0 | 69.7 / 50.4 | 48.0 / 24.5 | 25.4 / 5.7 |
| ST | **85.9** / 80.7 | 74.1 / 67.9 | 56.3 / 46.3 | 75.2 / 53.7 | 53.8 / 25.3 | 28.9 / 7.6 |
| Dreaming | **85.9** / 80.7 | **74.8 / 68.6** | **56.4 / 47.7** | **76.6 / 54.2** | **56.3 / 31.3** | **34.4 / 10.2** |
| SN only | **81.0** / 80.9 | 72.6 / 67.2 | 55.0 / 47.5 | **78.7 / 67.3** | 64.9 / 45.1 | 43.4 / 18.0 |
| SN + ST | 80.6 / 80.5 | 73.5 / 72.5 | 56.0 / 48.8 | 78.1 / 65.9 | 66.6 / 45.5 | 45.9 / 18.7 |
| SN + Dreaming | 80.5 / 80.3 | **74.3 / 72.9** | **56.6 / 49.7** | 78.1 / 65.9 | **67.0 / 48.5** | **46.7 / 22.0** |

(c) KITTI to nuScenes

| Range(m) | IoU 0.5 | | | IoU 0.7 | | |
|---|---|---|---|---|---|---|
| Method | 0-30 | 30-50 | 50-80 | 0-30 | 30-50 | 50-80 |
| in-domain | 65.6 / 64.5 | 18.4 / 17.2 | 3.5 / 2.8 | 57.8 / 37.0 | 15.8 / 5.3 | 2.6 / 0.8 |
| no retraining | 48.6 / 40.7 | 11.0 / 4.5 | 1.4 / **0.9** | 33.3 / 13.4 | 4.5 / 0.7 | 0.3 / **0.0** |
| ST | 52.2 / 43.9 | 11.8 / 9.1 | **9.1** / 0.4 | 33.2 / 8.6 | 9.1 / **4.5** | 3.0 / **0.0** |
| Dreaming | **53.6 / 45.8** | **13.8 / 9.7** | 4.5 / 0.4 | **39.1 / 14.2** | **9.8** / 3.0 | **4.5** / **0.0** |
| SN only | 52.9 / 47.0 | 11.1 / 10.0 | 1.0 / 0.4 | 44.7 / 22.0 | 10.2 / 4.5 | 0.6 / **0.1** |
| SN + ST | 51.9 / 47.3 | 11.5 / 10.1 | 1.4 / 0.6 | 45.6 / **26.3** | 10.6 / **9.1** | 0.9 / **0.1** |
| SN + Dreaming | **53.4 / 51.4** | **13.7 / 10.5** | **10.1 / 9.1** | **50.0** / 24.3 | **11.3** / **9.1** | **9.1** / **0.1** |

(d) KITTI to Waymo

| Range(m) | IoU 0.5 | | | IoU 0.7 | | |
|---|---|---|---|---|---|---|
| Method | 0-30 | 30-50 | 50-80 | 0-30 | 30-50 | 50-80 |
| in-domain | 81.6 / 81.5 | 71.7 / 71.2 | 58.2 / 50.8 | 80.8 / 68.8 | 69.1 / 54.9 | 47.0 / 27.2 |
| no retraining | 80.5 / 71.5 | 69.0 / 60.3 | 42.8 / 33.4 | 43.5 / 16.6 | 34.6 / 11.0 | 20.6 / **10.8** |
| ST | 81.0 / 78.4 | 69.5 / 61.4 | 48.5 / 39.3 | 48.2 / **18.6** | 36.8 / 16.0 | 23.6 / 7.0 |
| Dreaming | **81.1 / 78.5** | **69.9 / 61.8** | **50.2 / 41.0** | **51.4** / 13.8 | **44.5 / 16.7** | **25.6** / 7.8 |
| SN only | 81.0 / 80.4 | 70.0 / 62.4 | 42.7 / 40.7 | 71.4 / **53.4** | 60.9 / 39.8 | 38.0 / 19.4 |
| SN + ST | 81.1 / 80.6 | 70.5 / **68.5** | 49.7 / 42.5 | **78.1** / 53.1 | 61.7 / 45.1 | 40.2 / 20.6 |
| SN + Dreaming | **81.2 / 80.8** | **70.9** / 68.3 | **51.1 / 48.3** | **78.1** / 51.6 | **67.4 / 45.5** | **41.1 / 20.8** |

(e) Argoverse to KITTI

| Range(m) | IoU 0.5 | | | IoU 0.7 | | |
|---|---|---|---|---|---|---|
| Method | 0-30 | 30-50 | 50-80 | 0-30 | 30-50 | 50-80 |
| in-domain | 90.0 / 89.9 | 81.0 / 79.9 | 40.4 / 36.3 | 89.0 / 78.0 | 70.3 / 51.5 | 26.6 / 9.8 |
| no retraining | 89.4 / 88.8 | 71.0 / 65.2 | 18.0 / 13.9 | 72.6 / 47.8 | 35.8 / 14.6 | 4.9 / 3.0 |
| ST | **89.5 / 89.2** | 73.2 / 68.5 | 28.2 / 22.5 | 76.8 / 52.9 | 44.2 / 20.6 | 13.1 / 2.1 |
| Dreaming | 89.3 / **89.2** | **74.6 / 72.4** | **30.1 / 25.6** | **77.6 / 54.7** | **49.9 / 24.3** | **14.5 / 3.4** |
| SN only | 89.3 / 88.2 | 69.6 / 65.4 | 14.6 / 13.3 | 83.8 / 59.1 | 53.5 / 27.2 | 9.3 / 3.6 |
| SN + ST | **89.8** / 89.3 | 74.4 / 72.8 | 22.3 / 21.3 | **87.0** / 70.4 | 62.2 / 37.7 | 16.4 / 7.1 |
| SN + Dreaming | **89.8 / 89.4** | **75.4 / 73.8** | **29.4 / 25.4** | **87.0 / 73.5** | **62.8 / 41.9** | **17.2 / 10.3** |

(f) Argoverse to Lyft

| Range(m) | IoU 0.5 | | | IoU 0.7 | | |
|---|---|---|---|---|---|---|
| Method | 0-30 | 30-50 | 50-80 | 0-30 | 30-50 | 50-80 |
| in-domain | 89.2 / 88.9 | 78.4 / 77.9 | 67.2 / 65.5 | 88.4 / 79.0 | 74.8 / 54.6 | 54.7 / 27.3 |
| no retraining | 79.9 / 79.8 | 68.5 / 67.4 | 46.3 / 42.9 | 77.2 / 54.8 | 57.6 / 32.0 | 31.8 / 12.4 |
| ST | 80.2 / **80.1** | 72.7 / 67.9 | 48.8 / 46.0 | 78.3 / 55.4 | 63.3 / 35.8 | 35.0 / 11.3 |
| Dreaming | **80.3 / 80.1** | **73.8 / 72.4** | **54.5 / 47.6** | **78.9 / 57.3** | **64.2 / 37.1** | **35.6 / 12.8** |
| SN only | 79.6 / 79.2 | 67.1 / 66.0 | 44.8 / 41.8 | 75.7 / 54.7 | 55.3 / 30.4 | 32.4 / **14.2** |
| SN + ST | 80.1 / 79.9 | 72.2 / 67.4 | 47.4 / 45.3 | 78.3 / 56.0 | 63.6 / 38.0 | 36.4 / 13.5 |
| SN + Dreaming | **80.2 / 80.0** | **73.5 / 68.2** | **54.0 / 47.5** | **78.5 / 56.8** | **64.1 / 39.3** | **41.9** / 14.1 |

(g) Argoverse to nuScenes

| Range(m) | IoU 0.5 | | | IoU 0.7 | | |
|---|---|---|---|---|---|---|
| Method | 0-30 | 30-50 | 50-80 | 0-30 | 30-50 | 50-80 |
| in-domain | 65.6 / 64.5 | 18.4 / 17.2 | 3.5 / 2.8 | 57.8 / 37.0 | 15.8 / 5.3 | 2.6 / 0.8 |
| no retraining | **51.6 / 45.3** | **10.3 / 9.1** | 0.6 / 0.1 | **43.3** / 17.9 | **9.1** / 0.3 | 0.2 / **0.1** |
| ST | 47.7 / 43.4 | 6.3 / 5.4 | **9.1** / 0.2 | 41.5 / **18.4** | 5.6 / **4.5** | 0.3 / **0.1** |
| Dreaming | 48.1 / 43.5 | 8.0 / 4.5 | **3.0 / 3.0** | 41.4 / **18.4** | 5.3 / 2.3 | **3.0** / 0.0 |
| SN only | 47.6 / 45.6 | 5.7 / 4.5 | 0.5 / 0.2 | 43.1 / 18.2 | 4.5 / 0.7 | 0.1 / 0.0 |
| SN + ST | 49.1 / 44.5 | 11.0 / 4.2 | **9.1 / 9.1** | 42.7 / 22.4 | 10.4 / 1.8 | **9.1 / 9.1** |
| SN + Dreaming | **50.7 / 45.9** | **12.8 / 10.1** | **9.1 / 9.1** | **44.1 / 24.7** | **10.7 / 9.1** | **9.1** / 4.5 |

(h) Argoverse to Waymo

| Range(m) | IoU 0.5 | | | IoU 0.7 | | |
|---|---|---|---|---|---|---|
| Method | 0-30 | 30-50 | 50-80 | 0-30 | 30-50 | 50-80 |
| in-domain | 81.6 / 81.5 | 71.7 / 71.2 | 58.2 / 50.8 | 80.8 / 68.8 | 69.1 / 54.9 | 47.0 / 27.2 |
| no retraining | 81.2 / 80.4 | 69.3 / 61.7 | 50.4 / 47.8 | 70.8 / 43.7 | 59.6 / 35.0 | 39.1 / 18.8 |
| ST | **81.3 / 80.7** | 69.9 / **67.1** | 50.9 / 48.2 | **78.0 / 50.0** | **60.8** / 36.0 | 40.7 / **20.2** |
| Dreaming | **81.3** / 80.6 | **70.0** / 67.0 | **56.0 / 49.1** | 77.3 / 43.4 | 60.5 / **36.2** | **40.9 / 20.2** |
| SN only | 81.3 / 80.7 | 69.3 / 61.6 | 49.7 / 47.4 | 71.0 / 51.8 | 59.1 / 34.3 | 38.3 / 15.0 |
| SN + ST | **81.4 / 81.0** | 70.1 / **67.7** | 50.7 / 48.0 | **78.4 / 55.5** | **61.1** / 41.1 | 40.4 / **20.0** |
| SN + Dreaming | **81.4 / 81.0** | **70.4** / 67.5 | **56.1 / 49.5** | 77.9 / 53.6 | **61.1 / 41.8** | **44.9 / 20.0** |

(i) Lyft to KITTI

| Range(m) | IoU 0.5 | | | IoU 0.7 | | |
|---|---|---|---|---|---|---|
| Method | 0-30 | 30-50 | 50-80 | 0-30 | 30-50 | 50-80 |
| in-domain | 90.0 / 89.9 | 81.0 / 79.9 | 40.4 / 36.3 | 89.0 / 78.0 | 70.3 / 51.5 | 26.6 / 9.8 |
| no retraining | 88.9 / 88.6 | 67.7 / 64.9 | 26.1 / 20.4 | 65.2 / 38.7 | 36.1 / 12.9 | 8.5 / 2.2 |
| ST | 89.9 / 89.6 | 73.5 / 68.8 | 30.8 / 24.0 | 71.7 / 40.8 | 42.2 / **20.5** | 10.8 / 2.9 |
| Dreaming | **90.0 / 89.7** | **74.7 / 72.8** | **33.9 / 26.4** | **74.2 / 42.8** | **46.2** / 18.7 | **12.1 / 3.6** |
| SN only | 89.5 / 89.5 | 67.4 / 66.5 | 28.7 / 24.9 | 88.0 / 76.0 | 57.2 / 33.8 | 20.1 / 6.4 |
| SN + ST | 90.0 / 90.0 | 73.2 / 72.3 | 31.9 / 28.4 | 88.4 / 77.2 | 63.1 / 42.3 | 22.1 / 10.0 |
| SN + Dreaming | **90.1 / 90.1** | **76.2 / 74.8** | **36.1 / 32.1** | **88.6 / 78.0** | **64.6 / 42.5** | **23.6 / 10.7** |

(j) Lyft to Argoverse

| Range(m) | IoU 0.5 | | | IoU 0.7 | | |
|---|---|---|---|---|---|---|
| Method | 0-30 | 30-50 | 50-80 | 0-30 | 30-50 | 50-80 |
| in-domain | 88.6 / 85.7 | 68.5 / 66.4 | 37.8 / 30.0 | 76.5 / 53.2 | 56.6 / 30.4 | 20.2 / 10.1 |
| no retraining | 86.4 / 78.6 | 58.2 / 55.3 | 29.3 / 25.3 | 72.5 / 40.1 | 44.6 / 17.3 | 17.0 / **9.1** |
| ST | 86.7 / **84.2** | 63.6 / 57.0 | 30.7 / 26.7 | 74.8 / **42.3** | **50.4 / 19.2** | 18.1 / 4.5 |
| Dreaming | **87.3 / 84.1** | **64.9 / 61.3** | **35.2 / 27.3** | **75.1 / 41.9** | 49.7 / 18.7 | **18.6 / 9.1** |
| SN only | 86.5 / 78.4 | 58.1 / 55.0 | 29.2 / 22.2 | 72.3 / 46.5 | 44.3 / 13.0 | 17.7 / 9.1 |
| SN + ST | 86.4 / 78.6 | 63.8 / 56.9 | 30.6 / 26.9 | 74.8 / **48.6** | **50.8 / 21.4** | 18.9 / **10.1** |
| SN + Dreaming | **87.4 / 84.1** | **64.8 / 61.5** | **34.9 / 27.9** | **74.9** / 45.6 | 50.4 / 20.6 | **19.0** / 9.9 |

(k) Lyft to nuScenes

| Range(m) | IoU 0.5 | | | IoU 0.7 | | |
|---|---|---|---|---|---|---|
| Method | 0-30 | 30-50 | 50-80 | 0-30 | 30-50 | 50-80 |
| in-domain | 65.6 / 64.5 | 18.4 / 17.2 | 3.5 / 2.8 | 57.8 / 37.0 | 15.8 / 5.3 | 2.6 / 0.8 |
| no retraining | **53.2** / 47.1 | 9.4 / 3.8 | 4.5 / 0.4 | 45.9 / 24.0 | 7.3 / 0.9 | 4.5 / **0.3** |
| ST | 51.6 / 46.2 | 13.0 / **10.4** | **9.1 / 9.1** | 45.2 / 25.8 | 11.6 / **9.1** | **9.1** / 0.1 |
| Dreaming | 52.8 / **50.4** | **14.9** / 10.2 | 6.0 / 1.1 | **49.6 / 26.7** | **11.8 / 9.1** | 4.5 / 0.1 |
| SN only | **53.5** / 47.0 | 9.0 / 3.6 | 3.0 / 0.4 | 45.7 / 24.1 | 6.4 / 2.3 | 1.0 / 0.0 |
| SN + ST | 52.0 / 46.6 | 12.9 / **10.5** | 4.5 / **4.5** | 45.6 / **25.4** | 11.4 / **9.1** | **4.5 / 0.1** |
| SN + Dreaming | 52.3 / **49.8** | **15.1** / 10.2 | **5.0** / 1.5 | **48.8** / 23.3 | **12.1** / **9.1** | 3.0 / 0.0 |

(l) Lyft to Waymo

| Range(m) | IoU 0.5 | | | IoU 0.7 | | |
|---|---|---|---|---|---|---|
| Method | 0-30 | 30-50 | 50-80 | 0-30 | 30-50 | 50-80 |
| in-domain | 81.6 / 81.5 | 71.7 / 71.2 | 58.2 / 50.8 | 80.8 / 68.8 | 69.1 / 54.9 | 47.0 / 27.2 |
| no retraining | **81.5** / 81.2 | 70.9 / 69.8 | 51.6 / 49.6 | **79.5** / 55.8 | 61.6 / 44.5 | 40.5 / 18.6 |
| ST | **81.5** / 81.3 | 71.1 / 69.8 | 51.9 / 50.6 | 79.3 / **56.1** | **67.5 / 46.0** | **46.3 / 23.8** |
| Dreaming | **81.5** / 81.2 | **71.2 / 70.2** | **56.9 / 50.8** | 79.1 / 55.7 | 67.2 / 45.5 | 46.0 / 21.2 |
| SN only | 81.4 / 81.3 | 71.0 / 70.1 | 51.1 / 49.5 | 79.8 / **65.5** | 61.1 / 46.1 | 39.0 / 21.3 |
| SN + ST | 81.4 / 81.3 | 71.2 / 70.2 | 51.9 / 50.5 | **80.3** / 65.0 | **67.5 / 47.5** | **45.5 / 25.1** |
| SN + Dreaming | 81.4 / 81.3 | **71.3 / 70.4** | **56.9 / 50.9** | 80.1 / 64.9 | 67.1 / **47.5** | 44.6 / 22.8 |

(m) nuScenes to KITTI

| Range(m) | IoU 0.5 | | | IoU 0.7 | | |
|---|---|---|---|---|---|---|
| Method | 0-30 | 30-50 | 50-80 | 0-30 | 30-50 | 50-80 |
| in-domain | 90.0 / 89.9 | 81.0 / 79.9 | 40.4 / 36.3 | 89.0 / 78.0 | 70.3 / 51.5 | 26.6 / 9.8 |
| no retraining | 68.5 / 57.2 | 46.1 / 33.5 | 9.6 / 5.5 | 35.7 / 5.3 | 12.7 / 1.4 | 2.8 / **1.8** |
| ST | **87.8 / 78.9** | 64.8 / 51.0 | 16.4 / 8.7 | 45.9 / 10.9 | 21.4 / 2.5 | 3.4 / 0.6 |
| Dreaming | 87.6 / 78.8 | **69.3 / 56.3** | **21.7 / 13.4** | **46.4 / 14.5** | **25.7 / 10.2** | **5.4** / 0.5 |
| SN only | 78.8 / 78.3 | 48.9 / 46.6 | 6.0 / 5.5 | 74.9 / 40.9 | 37.7 / 11.4 | 4.7 / 1.1 |
| SN + ST | **88.7 / 88.5** | 65.6 / 63.5 | 16.6 / 13.0 | **84.3 / 61.2** | 53.3 / 22.3 | **10.6** / **1.8** |
| SN + Dreaming | 88.5 / 88.2 | **69.9 / 65.8** | **23.9 / 20.9** | 84.0 / 60.8 | **53.9 / 24.5** | 8.9 / 1.5 |

(n) nuScenes to Argoverse

| Range(m) | IoU 0.5 | | | IoU 0.7 | | |
|---|---|---|---|---|---|---|
| Method | 0-30 | 30-50 | 50-80 | 0-30 | 30-50 | 50-80 |
| in-domain | 88.6 / 85.7 | 68.5 / 66.4 | 37.8 / 30.0 | 76.5 / 53.2 | 56.6 / 30.4 | 20.2 / 10.1 |
| no retraining | 39.5 / 37.0 | 21.6 / 17.4 | 3.0 / 3.0 | 28.6 / 5.1 | 17.6 / **9.1** | 3.0 / 0.1 |
| ST | 65.2 / 59.5 | 32.6 / 25.9 | 12.4 / 9.1 | **51.9 / 15.7** | 24.3 / **9.1** | 9.1 / **0.6** |
| Dreaming | **65.9 / 63.2** | **50.0 / 40.3** | **17.7 / 13.0** | 50.7 / 10.6 | **32.3** / 2.9 | **9.6** / **0.6** |
| SN only | 49.6 / 47.8 | 20.8 / 16.4 | 4.5 / 4.5 | 38.2 / 8.2 | 16.6 / 2.3 | 4.5 / **1.8** |
| SN + ST | 56.7 / 55.5 | 23.3 / 18.0 | 4.7 / 3.4 | 50.3 / **13.3** | 17.4 / **9.1** | 3.3 / 0.5 |
| SN + Dreaming | **64.3 / 61.8** | **45.7 / 40.9** | **19.4 / 14.1** | **50.8** / 11.0 | **33.4** / 3.0 | **12.8** / 0.8 |

(o) nuScenes to Lyft

| Range(m) | IoU 0.5 | | | IoU 0.7 | | |
|---|---|---|---|---|---|---|
| Method | 0-30 | 30-50 | 50-80 | 0-30 | 30-50 | 50-80 |
| in-domain | 89.2 / 88.9 | 78.4 / 77.9 | 67.2 / 65.5 | 88.4 / 79.0 | 74.8 / 54.6 | 54.7 / 27.3 |
| no retraining | 52.3 / 51.0 | 41.1 / 37.9 | 22.9 / 15.1 | 49.5 / 17.9 | 36.8 / 10.5 | 15.1 / 9.1 |
| ST | **79.7** / 78.0 | 65.9 / 58.0 | 39.1 / 30.2 | **77.0** / 20.2 | 57.0 / **13.0** | 30.0 / **10.4** |
| Dreaming | 79.6 / **78.1** | **66.8 / 63.2** | **46.4 / 37.8** | **77.0** / 20.7 | **62.1** / 10.4 | **36.4** / 5.2 |
| SN only | 62.3 / 61.7 | 41.6 / 39.9 | 16.7 / 15.4 | 60.1 / 26.8 | 32.1 / 5.7 | 15.6 / 9.1 |
| SN + ST | 78.6 / 77.9 | 56.1 / 53.7 | 23.1 / 13.9 | **77.3 / 32.4** | 53.0 / 13.8 | 20.4 / 3.0 |
| SN + Dreaming | **78.7 / 78.0** | **64.7 / 57.5** | **39.7 / 36.1** | **77.3** / 31.6 | **56.0 / 16.0** | **29.6 / 11.0** |

(p) nuScenes to Waymo

| Range(m) | IoU 0.5 | | | IoU 0.7 | | |
|---|---|---|---|---|---|---|
| Method | 0-30 | 30-50 | 50-80 | 0-30 | 30-50 | 50-80 |
| in-domain | 81.6 / 81.5 | 71.7 / 71.2 | 58.2 / 50.8 | 80.8 / 68.8 | 69.1 / 54.9 | 47.0 / 27.2 |
| no retraining | 62.2 / 61.2 | 50.7 / 42.1 | 24.9 / 22.5 | 60.5 / 25.3 | 41.7 / 13.8 | 21.3 / 4.5 |
| ST | **80.7 / 71.9** | 68.7 / 60.9 | 42.5 / 39.6 | 71.6 / **34.5** | 60.7 / **24.6** | 37.0 / 8.8 |
| Dreaming | **80.7 / 71.9** | **69.4 / 61.3** | **50.1 / 45.5** | **71.7** / 33.5 | **61.2** / 24.3 | **39.3 / 10.7** |
| SN only | 71.1 / 70.3 | 50.6 / 41.8 | 25.1 / 16.6 | 61.1 / 34.8 | 40.5 / 14.4 | 16.1 / 2.8 |
| SN + ST | 80.7 / 79.1 | 67.4 / 59.2 | 41.3 / 32.1 | 71.4 / **44.0** | 58.2 / 27.3 | 29.7 / 7.4 |
| SN + Dreaming | **81.1 / 79.2** | **69.5 / 61.5** | **50.1 / 41.3** | **71.7** / 42.2 | **60.7 / 27.6** | **38.5 / 14.4** |

(q) Waymo to KITTI

| Range(m) | IoU 0.5 | | | IoU 0.7 | | |
|---|---|---|---|---|---|---|
| Method | 0-30 | 30-50 | 50-80 | 0-30 | 30-50 | 50-80 |
| in-domain | 90.0 / 89.9 | 81.0 / 79.9 | 40.4 / 36.3 | 89.0 / 78.0 | 70.3 / 51.5 | 26.6 / 9.8 |
| no retraining | 88.4 / 87.0 | 64.7 / 55.2 | 14.3 / 8.5 | 45.8 / 10.4 | 22.9 / 3.1 | 2.5 / 0.8 |
| ST | 89.3 / 88.3 | 70.5 / 62.8 | 25.6 / 18.1 | **50.7** / 10.1 | 28.6 / 11.9 | 4.3 / 2.3 |
| Dreaming | **89.5 / 88.7** | **72.6 / 65.8** | **27.8 / 19.7** | 49.8 / **16.3** | **31.9 / 12.3** | **5.0 / 3.0** |
| SN only | 88.6 / 88.5 | 63.0 / 61.9 | 11.6 / 10.7 | 84.1 / 53.8 | 51.7 / 28.3 | 8.4 / 5.1 |
| SN + ST | **89.7 / 89.7** | 72.5 / 69.0 | 19.8 / 18.6 | **87.1** / 60.6 | **60.8 / 38.6** | 13.8 / **6.2** |
| SN + Dreaming | 89.6 / 89.6 | **73.5 / 72.3** | **22.1 / 19.6** | 86.3 / **63.1** | 58.0 / 36.4 | **14.0** / 4.0 |

(r) Waymo to Argoverse

| Range(m) | IoU 0.5 | | | IoU 0.7 | | |
|---|---|---|---|---|---|---|
| Method | 0-30 | 30-50 | 50-80 | 0-30 | 30-50 | 50-80 |
| in-domain | 88.6 / 85.7 | 68.5 / 66.4 | 37.8 / 30.0 | 76.5 / 53.2 | 56.6 / 30.4 | 20.2 / 10.1 |
| no retraining | 84.0 / 76.3 | 54.4 / 51.3 | 24.7 / 19.6 | 69.6 / 28.4 | 40.6 / 13.8 | 15.9 / 1.6 |
| ST | 85.7 / **78.6** | 55.5 / 52.9 | 23.4 / 21.3 | **70.7 / 29.8** | **44.3 / 14.0** | 14.9 / 2.1 |
| Dreaming | **85.8** / 78.3 | **56.7 / 54.6** | **28.3 / 22.3** | 64.8 / 28.8 | 44.1 / 13.5 | **17.3 / 3.0** |
| SN only | 83.3 / 74.9 | 54.3 / 47.2 | 19.2 / 16.3 | 69.3 / 29.2 | 43.2 / 16.9 | 10.3 / 3.0 |
| SN + ST | 85.2 / 78.0 | 55.4 / 52.8 | 22.8 / 20.4 | **73.9 / 37.6** | **45.8 / 19.8** | 14.8 / **9.1** |
| SN + Dreaming | **85.4 / 78.1** | **56.0 / 53.6** | **27.7 / 22.1** | 71.8 / 35.2 | 45.5 / 16.8 | **17.6** / 9.1 |

(s) Waymo to Lyft

| Range(m) | IoU 0.5 | | | IoU 0.7 | | |
|---|---|---|---|---|---|---|
| Method | 0-30 | 30-50 | 50-80 | 0-30 | 30-50 | 50-80 |
| in-domain | 89.2 / 88.9 | 78.4 / 77.9 | 67.2 / 65.5 | 88.4 / 79.0 | 74.8 / 54.6 | 54.7 / 27.3 |
| no retraining | 80.8 / **80.8** | 67.8 / 66.8 | 45.3 / 43.0 | 78.3 / 55.3 | 56.7 / 31.2 | 34.4 / 10.5 |
| ST | 80.7 / 80.6 | 74.6 / 73.4 | 54.0 / 48.1 | 78.9 / 54.9 | **65.4 / 32.8** | 42.9 / 17.2 |
| Dreaming | **86.8 / 80.8** | **75.4 / 74.0** | **55.5 / 49.6** | **79.2 / 55.5** | **65.4** / 32.6 | **44.6 / 17.3** |
| SN only | 80.5 / 80.2 | 66.4 / 65.1 | 38.4 / 37.1 | 78.0 / 55.9 | 55.8 / 31.4 | 33.5 / 15.4 |
| SN + ST | 86.8 / 80.7 | 74.2 / 73.3 | 45.5 / 44.7 | 79.8 / 63.9 | 65.4 / **38.1** | 36.1 / 14.5 |
| SN + Dreaming | **87.3 / 81.1** | **75.3 / 73.8** | **48.0 / 47.0** | **80.3 / 64.6** | **66.3 / 38.1** | **38.4 / 18.4** |

(t) Waymo to nuScenes

| Range(m) | IoU 0.5 | | | IoU 0.7 | | |
|---|---|---|---|---|---|---|
| Method | 0-30 | 30-50 | 50-80 | 0-30 | 30-50 | 50-80 |
| in-domain | 65.6 / 64.5 | 18.4 / 17.2 | 3.5 / 2.8 | 57.8 / 37.0 | 15.8 / 5.3 | 2.6 / 0.8 |
| no retraining | 46.4 / 43.2 | 3.0 / 3.0 | 0.2 / 0.0 | 42.6 / 23.9 | 3.0 / 0.2 | 0.1 / 0.0 |
| ST | 51.4 / 46.5 | 10.7 / 9.1 | 3.0 / 0.3 | 45.8 / **26.4** | 9.1 / **3.0** | 3.0 / 0.0 |
| Dreaming | **53.8 / 48.0** | **12.4 / 10.0** | **9.1 / 9.1** | **47.4** / 23.7 | **11.1** / 3.0 | **9.1** / 0.8 |
| SN only | 45.8 / 43.5 | 9.1 / 9.1 | 0.1 / 0.0 | 42.9 / 23.5 | 9.1 / 0.2 | 0.0 / **0.0** |
| SN + ST | 51.0 / 45.8 | 9.1 / 9.1 | 1.0 / 0.3 | 44.7 / 25.6 | 9.1 / **9.1** | 1.0 / **0.0** |
| SN + Dreaming | **53.7 / 47.6** | **12.4 / 10.2** | **2.8** / 0.8 | **46.8 / 26.5** | **11.2** / 2.3 | **2.3** / 0.0 |

