# OpenReview forum: "Exploiting Playbacks in Unsupervised Domain Adaptation for 3D Object Detection"
_ICLR.cc/2021/Conference — Reject_

### Official Review · AnonReviewer4 · 2020-10-24
**This paper addresses the ability of 3D object detection within the context of self-driving cars to adapt to new unlabeled data by using extended kalman filter in temporal sequences.**

**Rating:** 6
**Confidence:** 4

**Review:**

Pros:
- The proposed method is simple yet effective and has wide uses in real-world applications
- Solid experiments across 5 benchmarks.
- This method does not rely on the source domain data and learned trackers.


Cons:
- The object detector will detect objects accurately only when they are close to the self-driving car. The claim is not supported when there is a large domain gap (e.g., different LiDARs or significantly different scenarios). The proposed model will fail to handle this situation.
- For the static cars, why don't use ego-motion to model the temporal relationships? It should have a better performance than EKF.
- The generation of the pseudo-labels depends heavily on the confidence scores obtained from the object detector. Confidence scores > 0.8. How is the threshold of 0.80 chosen? Would other thresholds be more effective?
- Why does the author only post results in 50-80m in Tab. 3. The accurate detection in 0-50m is more important, although the relative improvement may be less.
- The method is somewhat similar to the existing tracker-based UDA methods, thus the novelty is limited. However, the application of 3D detection and the extensive experiments are great and may benefit further research significantly.

---

> ### Author Response · Authors · 2020-11-18
> **Thanks for your comments!**
>
> We thank the reviewer for the constructive comments! We address your concerns as follows:
> 1. **Novelty / Contribution**: please see our general comments above.
> 2. **The object detector will detect objects accurately only when they are close to the self-driving car**:
> It is true that if the sensors were to change drastically between source and target domains, one could imagine a setting where a car suddenly detects nothing - making it impossible to improve through our method. However, we do not observe this in any of our data sets. As our main focus is to reduce the adaptation gap, we believe that it is fair to assume the detector achieves a minimal level of accuracy even in the target domain.
> 3. **static cars**: Thanks for pointing out! You are correct, and we do utilize the ego-motion in the EKF via the physics based model (relative between two cars). Thus, this temporal motion is captured with static vehicles. We will make this clearer in the paper.
> 4. **How 0.8 threshold is chosen**: We follow the common practice of DA to select the hyper-parameters on a single source-and-target pair (from Argoverse to KITTI). We select the threshold by optimizing the performance of the **baseline ST** on the target train set (after ST, we reveal true labels) to get a good balance between close and far ranges. We then apply the same threshold to all the experiments.
> Results are shown below. It can be seen that by choosing the threshold 0.8 the ST method can have a good balance between close and farther ranges.
> | AP_BEV / AP_3D | IoU 0.5     |             |             | IoU 0.7     |             |            |
> |----------------|-------------|-------------|-------------|-------------|-------------|------------|
> | range          |     0-30    |    30-50    |    50-80    |     0-30    |    30-50    |    50-80   |
> | ST 0.7         | 80.7 / 80.5 | 69.6 / 65.2 | 31.0 / 24.7 | 65.9 / 41.4 | 37.9 / 13.1 | 16.4 / 4.5 |
> | ST 0.8         | 80.7 / 80.4 | 69.1 / 64.7 | 30.9 / 24.3 | 66.1 / 40.3 | 38.0 / 13.0 | 12.9 / 1.3 |
> | ST 0.9         | 80.7 / 80.5 | 70.0 / 65.8 | 29.4 / 24.8 | 66.4 / 41.6 | 40.0 / 13.1 | 12.7 / 1.8 |
> 5. **Results on other ranges**: We report all results on other ranges across 5 datasets in the appendix (see Table 8). We can see a consistent trend that on the close range (0-30m, 30-50m) our method is still better or on-par with the baselines, and on the farther range (50-80m) the improvement is most significant as expected. Due to the page limit of the main paper, we decided to show the part where our method improves most.

---

### Official Review · AnonReviewer1 · 2020-10-28
**The paper leverages two ideas for adapting 3d object detection to new domains: 1) self-training with pseudo labels and 2) using object tracking to refine and extend the initial pseudo labels by an object detector. The technical novelty is rather limited as both components have been widely used in the computer vision field. However, the paper is nicely written with detailed experiments.**

**Rating:** 6
**Confidence:** 5

**Review:**

The topic of adapting 3d object detectors to new domains is important. The paper clearly motivates the problem, clearly presents the methods and shows detailed experiments.  I really enjoyed reading the paper.

My main concern is that the two components of the method (self-training with pseudo labels and generating more pseudo labels with an object tracker for object detection) have been developed and widely used in the computer vision domain for 2d object detectors. The main novelty of this paper lies in using the counterparts of the two components in 3d for the new 3d object detection task. The use of self-training is almost the same as all previous methods. There are a few interesting engineering parts in using 3d object tracker to expand the pseudo labels such as label extrapolation and interpolation.

Another question I have is that when the object detector gets stronger, do we need a stronger object tracking algorithm in order to provide additional useful information. If the tracking algorithm is too weak, relative to object detection methods, the augmented pseudo labels will be too noisy to provide any help. Discussions or experiments on this point would be very helpful in understanding the application domain of the proposed method.

Although the novelty of the method is rather small, the authors have made good efforts in supporting the work with extensive experiments. The authors have evaluated their method on five datasets (all the 2 out of 5 combinations). The results are good across all the scenarios. The paper is clearly written and the method is well motivated.

I am not sure whether a paper with extensive experiments and relatively small technical contribution should be considered as a good paper for ICLR.  After reading other reviews and the rebuttal, I opt for acceptance.

---

> ### Author Response · Authors · 2020-11-18
> **Thanks for your comments!**
>
> We thank the reviewer for the constructive comments! We address your concerns as follows:
> 1. **Novelty / Contribution**: please see our general comments above.
> 2. **Stronger object tracking algorithm**: In general, a better tracker will always lead to improved adaptation results. In addition, a better detector will also always improve tracking results. Our approach is primarily improving the limits of the detector in terms of range and adaptation to new environments, which will always exist. We therefore expect our approach to help even as detectors (and trackers)  improve. We will clarify this in the final version.

---

### Official Review · AnonReviewer3 · 2020-10-29
**Paper1349**

**Rating:** 6
**Confidence:** 3

**Review:**


This paper proposed an unsupervised domain adaptation method for 3D lidar-based object detection. The idea is simple and straightforward: using cross-domain detector + offline tracking to provide pseudo-labels, inspired by similar UDA efforts for 2D detection. Experiments are conducted over multiple self-driving perception datasets, and results validated the effectiveness of the proposed method.

Pros:
* The idea is simple and straightforward. The approach is technically sound.
* The presentation is clear. The introduction is easy to follow and enjoyable to read; Related work is thorough properly reflects the current states. Technical details are clearly described so that reproducing should not be very difficult.
* The experiment showcases solid performance improvement over baselines (self-training and statistical normalization)
* The paper also conducted very detailed and convincing ablation studies.
* Consistent improvements have been seen in several datasets and two different detectors.

Cons:
* I have some concerns regarding claimed contributions/novelty.
* Offline tracking is not adequately benchmarked to justify the choice of extrapolation.
* The online tracker is not on par with the current state of the art.
* There is not enough pseudo GT quality analysis against manually labeled ground-truth.

The usage of "video" to produce confident pseudo labels for unsupervised domain adaptation has been stressed in the introduction. However, as the related work described, this has been explored before with a similar technical approach for 2D detection (offline tracking to produce labels); see Roy-Chowdhury et al.

It's hard to say if extrapolation is a significant contribution unless adequately benchmarked, showcasing the offline tracker has improved using this trick. Such benchmarking could be done on the KITTI tracking benchmark to compare with/without extrapolation procedure. The current ablation on UDA tells little information as improvement is not significant.

There is no comparison against other trackers. Based on the reported numbers, the online tracker adapted from Diaz-Ruiz et al. 2019 is subpar from the current state-of-the-art Kalman-based online tracker. It's hard to justify why this one is chosen. Why not Weng et al. 2020 or Chiu et al. 2020, as mentioned in the paper? In particular, the tracker in Weng et al. 2020 is open-sourced.

Please provide an mAP evaluation of the pseudo GT quality over some sequences with GT labels.

Although not required, it would be great to see whether the author plan to release the code.

--------------------------------------------------------- Post-rebuttal comments ---------------------------------------------------------------------------

I carefully read the rebuttal and other reviewer comments. The author addressed my concerns on pseudo-label quality assessment and comparison against SOTA trackers. From the experimental perspective, I am very convinced the paper did a great job now. Please incorporate these additional experiments into the paper making it more complete.

That being said, similar to other reviewers, I am not very convinced about the author's reply on novelty/contribution. It's true it has not been applied in 3D, which is new. However, I am not convinced by the claims in rebuttal, such as "using physics-based dynamics models" (I think you are referring to kinematics-based instead of physics-based), "3D extrapolations" (which could induce potential problems due to the multi-modal future uncertainty), and "self-training" (which is not new). Thus, if the paper gets accepted, I strongly encourage the author to rewrite the introduction and properly reflect the core contributions.

Overall I am still on the positive side. But I am fine with both decisions.

---

> ### Author Response · Authors · 2020-11-18
> **Thanks for your comments!**
>
> We thank the reviewer for the constructive comments! We address your concerns as follows:
> 1. **Novelty / Contribution**: please see our general comments above.
> 2. **pseudo-label quality**: Thanks for bringing this up! We did an evaluation on the pseudo-label quality under the Argo -> KITTI setting (see the table below). We show on the train set of KITTI the AP_BEV and AP_3D of pure pseudo-labels and pseudo-labels after smoothing, resizing, interpolation and extrapolation. It can be seen that the process significantly improves the label quality on all ranges. We will add this in the camera-ready version.
> Argo -> KITTI pseudo-label evaluation results:
> | AP_BEV / AP_3D     | IoU 0.5     |             |             |             | IoU 0.7     |             |            |             |
> |--------------------|-------------|-------------|-------------|-------------|-------------|-------------|------------|-------------|
> | range              | 0-30        | 30-50       | 50-80       | 0-80        | 0-30        | 30-50       | 50-80      | 0-80        |
> | PL                 | 80.5 / 80.0 | 63.7 / 60.7 | 23.9 / 18.6 | 69.0 / 67.3 | 63.9 / 36.6 | 33.0 / 12.1 | 10.0 / 0.9 | 46.6 / 24.2 |
> | PL + S + R + I + E | 80.5 / 80.2 | 68.0 / 61.1 | 29.5 / 22.6 | 73.9 / 67.5 | 64.9 / 38.0 | 38.4 / 15.1 | 12.1 / 0.9 | 51.1 / 25.0 |
>
> 3. **Benchmarking extrapolation and tracker**: Thanks for pointing out! We apologize that we did not make it clear, but we show the ablation with and without extrapolation in Table 5, and there are significant improvements in the 50-80 m range, where extrapolations are expected to help the most. Also, we did experiment with other trackers, including Weng et al. 2020, and found the Diaz-Ruiz tracker to be on-par or better, see the table below. We will clarify this in the final version. It may however be important to emphasize that our approach is tracker agnostic (provided it is not learned, as this may introduce another adaptation gap), and as trackers improve, so will our approach.
> In the NuScenes 2 KITTI setting (Evaluation on 11 sequences of Validation Set). Please refer to Weng et. al. where the tracking metrics are explained, but sAMOTA (higher is better) demonstrates that more vehicles are tracked properly and AMOTP (higher is better) demonstrates higher precision in localization.
> | Tracking Algorithm           | sAMOTA | AMOTA  | AMOTP  |
> |------------------------------|--------|--------|--------|
> | AB3DMOT (Wang et al)         | 0.1993 | 0.0279 | 0.1645 |
> | Ours (without extrapolation) | 0.3385 | 0.0777 | 0.2703 |
> | Ours (with extrapolation)    | 0.3743 | 0.0898 | 0.3501 |
> 4. **Code** Yes. We will definitely release the code upon acceptance.

---

### Official Review · AnonReviewer2 · 2020-10-29
**Practical but lack of novelty**

**Rating:** 4
**Confidence:** 4

**Review:**

The paper proposes the use of playbacks for UDA. The author uses the trained model and an offline 3D object tracker to generate high-quality pseudo-labels of the target domain. After that, the original model is fine-tuned on the generated pseudo-labels to improve performance on the target domain.

The paper can be understood in general and the writing is easy to follow. The results of the paper are practical. It's reasonable because it can generate more accurate 3D boxes for the target domain, especially for those long-distance objects. The authors have done experiments on 5 data sets to show the generalization capability of the method and compare the two decent baselines with the proposed method.

With the video of the same scene over time, I am also curious if the effect of using the point cloud of the previous/next frames to enhance the point cloud data of the current frame, which may further enhance the effect and generalization ability of pseudo-labels.

The major novelty of the paper is the combination of offline-tracking and self-training techniques, which is practical for real-world engineering problems. However, in general, I think the novelty is still limited to the ICLR community. In my view, the only difference between the proposed method and ST is the introduction of video information (an assumption) and the offline tracker to make the pseudo-label more accurate.

---

> ### Author Response · Authors · 2020-11-18
> **Thanks for your comments!**
>
> We thank the reviewer for the constructive comments! We address your concerns as follows:
> 1. **Novelty / Contribution**: please see our general comments above.
> 2. **Using the previous/next frames to enhance**: Yes we believe this is a very interesting direction and worthy of further investigation. More specifically, we plan to integrate and test the concept to see if the results further improve the current pipeline. However, we also believe that applying multiple frame-based detectors is a complementary idea to our setting, and does not fundamentally change our proposed novelty. Our focus is on DA for one frame-based detector without accessing the source data, but one can also study DA for video-based detectors in the same setting and our DA techniques (Dreaming) can still be applied.
> Nevertheless, we have performed a preliminary experiment where we trained a model, PIXOR, in KITTI and tested on KITTI both using previous and next frames and a single frame.  We found there was not a significant performance difference. We conjecture for moving vehicles this might not have a significant effect as the frame rate might be too low. This is a good idea, we will continue to investigate, and will add on the camera-ready.
> |AP_BEV | BEV IoU 0.5 |          |      | BEV IoU 0.7 |          |      |
> |--------------------|-------------|----------|------|-------------|----------|------|
> |                    |     Easy    | Moderate | Hard |     Easy    | Moderate | Hard |
> | Single-frame PIXOR |     90.0    |   80.5   | 80.0 |     87.5    |   77.4   | 77.2 |
> |  Three-frame PIXOR |     89.1    |   79.9   | 79.6 |     86.1    |   76.3   | 69.4 |

---

### Author Response · Authors · 2020-11-18
**General response in terms of novelty**

Multiple reviewers made statements along the lines that our submission is practical but lacks novelty. Although we agree that the architecture itself does not introduce novel components, we believe that the setting itself, and the insight of using physics-based dynamics models for 3D extrapolations and self-training, are very novel and have not been published previously. Adapting 3D detectors across domains is an extremely challenging problem, and we are the first to show large improvements in accuracy with no additional supervision --- essentially tripling accuracy in the 50-80 m range.

The fact that our approach is so simple makes this even more impressive and generally applicable to a broad range of scenarios. As reviewers would no doubt agree, sometimes novel, impactful conclusions take the form of identifying simple ideas and showing that they are surprisingly effective. We believe our paper provides such a simple but effective solution. Adding any additional and more complicated components (even if they could lead to slight improvements), would have been misleading and obscured this important message: that self-training with physics-based tracking alone is effective for adapting 3D object detectors to widely different domains.

---

### Decision · Program_Chairs · 2021-01-07
**Final Decision**

**Decision:**

Reject

**Comment:**

This paper proposed an unsupervised domain adaptation method for 3D lidar-based object detection. Four reviewers provided detailed reviews: 3 rated “Marginally above acceptance threshold”, and 1 rated “Ok but not good enough - rejection”. The reviewers appreciated simple yet effective idea, the well motivated method, the comprehensiveness of the experiments, and well written paper. However, major concerns are also raised regarding the core technical contributions on the proposed approach. The ACs look at the paper, the review, the rebuttal, and the discussion. Given the concerns on the core technical contributions, the high competitiveness of the ICLR field, and the lack of enthusiastic endorsements from reviewers, the ACs believe this work is not ready to be accepted to ICLR yet and hence a rejection decision is recommended.